# Ground-based contrail observations: comparisons with reanalysis weather and contrail model simulations

Jade Low[1], Roger Teoh[1], Joel Ponsonby[1], Edward Gryspeerdt[2], Marc Shapiro[3], and Marc E.J. Stettler[1]

[1]Department of Civil and Environmental Engineering, Imperial College London, London, SW7 2AZ, United Kingdom
[2]Grantham Institute for Climate Change and the Environment, Imperial College London, London, SW7 2AZ, United Kingdom
[3]Breakthrough Energy, 4110 Carillon Point, Kirkland, WA 98033, United States

*Correspondence to*: Marc E.J. Stettler (m.stettler@imperial.ac.uk)

**Abstract.** Observations of contrails are vital for improving our understanding of contrail formation and lifecycle, informing models, and assessing mitigation strategies. Here, we developed a methodology that utilises ground-based cameras for tracking and analysing young contrails (< 35 minutes) formed under clear sky conditions, comparing these observations against reanalysis meteorology and simulations from the contrail cirrus prediction model (CoCiP) with actual flight trajectories. Our observations consist of 14 h of video footage recorded over five different days in Central London, capturing 1,582 flight waypoints from 281 flights. The simulation correctly predicted contrail formation and absence for around 75% of these waypoints, with incorrect contrail predictions occurring at warmer temperatures than those with true positive predictions (7.8 K vs. 12.8 K below the Schmidt-Appleman Criterion threshold temperature). When evaluating contrails with observed lifetimes of at least 2 minutes, the simulation's correct prediction rate for contrail formation increases to over 85%. Among all waypoints with contrail observations, 78% of short-lived contrails (observed lifetimes < 2 minutes) formed under ice sub-saturated conditions, while 75% of persistent contrails (observed lifetimes > 10 minutes) formed under ice supersaturated conditions. On average, the simulated contrail geometric width was around 100 m smaller than the observed (visible) width over its observed lifetime, with the mean underestimation reaching up to 280 m within the first five minutes. Discrepancies between the observed and simulated contrail formation, lifetime and widths can be associated with uncertainties in reanalysis meteorology due to known model limitations and sub-grid scale variabilities, contrail model simplifications, uncertainties in aircraft performance estimates, and observational challenges, among other possible factors. Overall, this study demonstrates the potential of ground-based cameras to create essential observational and benchmark datasets for validating and improving existing weather and contrail models.

## 1 Introduction

Contrails form behind an aircraft at altitudes of 8–13 km when conditions in the exhaust plume fulfil the Schmidt-Appleman Criterion (SAC) (Schumann, 1996). Under these conditions, the relative humidity in the exhaust plume exceeds liquid saturation enabling water vapour to condense onto the surface of soot particles to form water droplets, which subsequently

freeze to form contrail ice crystals. These newly formed contrail ice particles are entrained in the aircraft's wake vortices, and in most cases, contrails that are formed disappear within a few minutes as adiabatic heating causes the ice particles to sublimate (Lewellen and Lewellen, 2001; Unterstrasser, 2016). However, a small fraction of contrails can persist beyond a few minutes when the atmosphere is ice supersaturated, i.e., relative humidity with respect to ice (RHi) exceeding 100% (Jensen et al., 1998a). According to the definition provided by the World Meteorological Organization (2017), contrails that survive for at least 10 minutes are known as persistent contrails. Over time, persistent contrails tend to spread and mix with other contrails and natural clouds to form contrail cirrus clusters (Haywood et al., 2009) affecting the Earth's radiative balance and producing a net warming effect (Fuglestvedt et al., 2010; Meerkötter et al., 1999). Recent studies suggest that the global annual mean contrail cirrus net radiative forcing (RF) in 2018 and 2019 (best-estimate of between 61 and 72 mW m$^{-2}$ across three studies) (Bier and Burkhardt, 2022a; Märkl et al., 2024; Quaas et al., 2021; Teoh et al., 2024a) could be around two times greater than the RF from aviation's cumulative $CO_2$ emissions (34.3 [31, 38] mW m$^{-2}$ at a 95% confidence interval) (Lee et al., 2021).

Different modelling approaches are available to simulate the contrail properties and climate forcing, including: (i) large-eddy simulations (LES) (Lewellen, 2014; Lewellen et al., 2014; Unterstrasser, 2016); (ii) general circulation models (GCM) (Bier and Burkhardt, 2022b; Chen and Gettelman, 2013; Märkl et al., 2024); (iii) Lagrangian models based on parameterised physics, such as the contrail cirrus prediction model (CoCiP) (Schumann, 2012); and (iv) climate change functions (CCFs) and algorithmic climate change functions (aCCFs) (Dietmüller et al., 2023; Grewe et al., 2014). These contrail modelling approaches have been used to estimate the global and regional contrail climate forcing (Bier and Burkhardt, 2022b; Chen and Gettelman, 2013; Schumann et al., 2021; Teoh et al., 2022a, 2024a) and explore the effectiveness of different mitigation strategies (Burkhardt et al., 2018; Caiazzo et al., 2017; Grewe et al., 2017; Märkl et al., 2024; Martin Frias et al., 2024; Schumann et al., 2011; Teoh et al., 2020, 2022b).

To enhance confidence and ensure that any proposed contrail mitigation solution yields a net climate benefit, it is crucial that these contrail models are extensively validated against measurements and observations. Existing studies have compared the simulated contrail properties from CoCiP relative to in-situ measurements, remote sensing data, and satellite observations, and generally found a good agreement between the measured and simulated contrail properties at various stages of their lifecycle (Jeßberger et al., 2013; Märkl et al., 2024; Schumann et al., 2017, 2021; Teoh et al., 2024a). However, these studies either focused on aggregate statistics derived from an ensemble of contrails or assessed the simulated contrail properties with in-situ measurements of young contrails at a single point in time with a limited number of data points. While satellite observations can partially address some limitations of in-situ measurements by enabling a large number of contrails to be measured, matched with specific flights and tracked over time (Duda et al., 2019; Gryspeerdt et al., 2024; Iwabuchi et al., 2012; Marjani et al., 2022; Tesche et al., 2016; Vázquez-Navarro et al., 2015), they still face challenges in detecting young contrails with sub-pixel width, aged contrail cirrus that has lost its line-shaped structure, instances of cloud-contrail overlap, and contrails with small optical depths (< 0.05) (Kärcher et al., 2010; Mannstein et al., 2010; Meijer et al., 2022).

Ground-based instruments, such as lidar and cameras, can complement in-situ measurements and satellite observations in validating contrail models (Mannstein et al., 2010; Rosenow et al., 2023; Schumann et al., 2013). Notably, contrail

observations from ground-based cameras can provide specific advantages over satellites, particularly in observing contrail formation and the early stages of their lifecycle, and detecting optically thin contrails (Mannstein et al., 2010). However, previous research using ground-based instruments has predominantly focused on natural cirrus observations (Feister et al., 2010; Long et al., 2006; Seiz et al., 2007), with only two small-scale studies comparing a total of 16 observed contrail properties (e.g., 3D positions, width, and/or persistence) with model estimates (Rosenow et al., 2023; Schumann et al., 2013).

Recognising the potential of ground-based cameras, this study aims to: (i) develop a methodology for detecting and tracking contrails over time and extracting their widths from ground-based camera footage; and (ii) evaluate these contrail observations against CoCiP simulations, which are informed by meteorological data from a reanalysis numerical weather prediction (NWP) model, on a larger scale than prior studies.

## 2 Materials and methods

This section describes the contrail observations provided by the ground-based camera (Section 2.1), the workflow that is used to simulate the formation and evolution of contrails (Section 2.2), and the methods used to superimpose the actual flight trajectories and simulated contrails onto the video footage (Section 2.3) and to compare between the observed and simulated contrails properties (Section 2.4). Figure 1 provides an overview of the step-by-step process and datasets used to compare the ground-based contrail observations with the simulated contrail outputs.

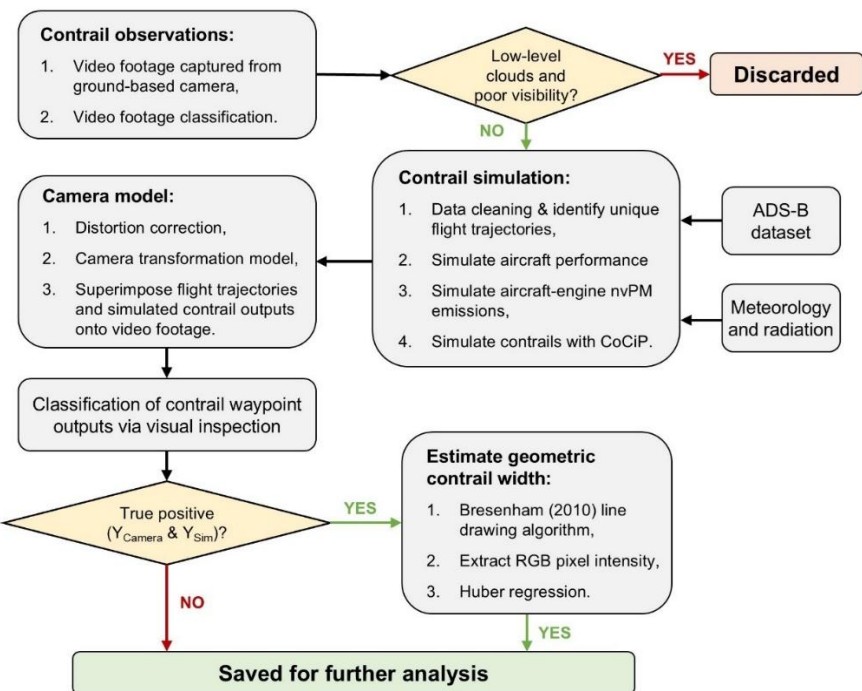

**Figure 1: Overview of the step-by-step process and datasets used to compare the ground-based contrail observations with the simulated contrail outputs from CoCiP.**

### 2.1 Contrail observations

Contrail observations were made using a Raspberry Pi Camera Module v2.1 which features an 8 Megapixel sensor (3280 x 2464 pixels), a wide-angle field of view spanning 62.2° horizontally and 48.8° vertically, and a focal length of 3.04 mm (Raspberry Pi, n.d.). The camera was positioned at Imperial College London's South Kensington Campus (51.4988°N, 0.1788°W) at an elevation of 25 m and pitched at a 25° angle above the horizontal plane. Recordings were taken between October-2021 and April-2022 during daylight hours, and at a temporal resolution of 5 seconds per frame. The captured footage is then filtered to remove the time intervals with low-level clouds and poor visibility (i.e., nighttime and periods with significant glare from direct sunlight) (Appendix A1). This filtering results in a final dataset containing 14 h of video footage collected over 5 different days.

### 2.2 Contrail simulation

The formation and evolution of contrails that were observed by the video footage are simulated using CoCiP (Schumann, 2012). For this study, we use the CoCiP algorithm hosted in the open-source pycontrails repository v0.52.2 (Shapiro et al., 2024). Several datasets and methods are required as inputs to CoCiP, including the: (i) actual flight trajectories; (ii) historical meteorology and radiation fields; and (iii) aircraft performance and emissions estimates.

#### 2.2.1 Flight trajectories and waypoint properties

The trajectories for each flight were derived using the raw Automatic Dependent Surveillance – Broadcast (ADS-B) telemetry that was purchased from Spire Aviation (Teoh et al., 2024b). Each ADS-B waypoint contains the unique flight identifier (call sign and flight number) and its corresponding 4D position (longitude, latitude, barometric altitude, and time) provided at time intervals of 40 s, and we filter the dataset to only include waypoints within a defined spatial bounding box (40 – 60° N and 10° W – 10° E) that extends ±10° in longitude and latitude from camera's location.

The Base of Aircraft Data Family 4.2 (BADA 4) aircraft performance model (EUROCONTROL, 2016) is used to estimate the: (i) fuel mass flow rate; (ii) change in aircraft mass, assuming that the initial aircraft mass at the first known waypoint is set to the nominal mass provided by BADA; and (iii) overall efficiency ($\eta$). The aircraft-engine specific non-volatile particulate matter (nvPM) number emissions index ($EI_n$), which strongly influences the initial contrail ice crystal properties, is estimated by interpolating the engine-specific nvPM emissions profile from the ICAO Aircraft Engine Emissions Databank (EDB) (EASA, 2021) relative to the non-dimensional engine thrust settings (Teoh et al., 2024b). All flights are assumed to be powered by conventional Jet A-1 fuel.

#### 2.2.2 Meteorology

The historical 4D meteorological fields within the defined spatial bounding box (between 40 – 60° N and 10° W – 10° E) were provided by the European Centre for Medium Range Weather Forecast (ECMWF) ERA5 high-resolution realisation (HRES)

reanalysis (ECMWF, 2021; Hersbach et al., 2020) at a spatial resolution of 0.25° longitude × 0.25° latitude over 37 pressure levels and at a 1 h temporal resolution. For each flight waypoint, the local meteorology is estimated from a quadrilinear interpolation across the three space coordinates and time (Schumann, 2012).

We apply the humidity correction methodology from Teoh et al. (2022a) to ensure that the ERA5-derived RHi has a probability density function that is consistent with in-situ measurements from the In-service Aircraft for a Global Observing System (IAGOS) dataset (Boulanger et al., 2022; Petzold et al., 2015),

$$\text{RHi}_{\text{corrected}} = \begin{cases} \frac{\text{RHi}}{a_{\text{opt}}} & \text{for } \left(\frac{\text{RHi}}{a_{\text{opt}}}\right) \le 1 \\ \min\left(\left(\frac{\text{RHi}}{a_{\text{opt}}}\right)^{b_{\text{opt}}}, \text{RHi}_{\text{max}}\right) & \text{for } \left(\frac{\text{RHi}}{a_{\text{opt}}}\right) > 1 \end{cases}, \tag{1}$$

where $\text{RHi}_{\text{max}} = 1.65$, $a_{\text{opt}} = 0.9779$ and $b_{\text{opt}} = 1.635$. Eq. (1) is expected to be applicable to this study because its coefficients were calibrated using RHi measurements over the North Atlantic (40 – 75° N and 50 – 10° W), which corresponds to the same latitude band as our study domain (40 – 60° N and 10° W – 10° E). While Eq. (1) improves the goodness of fit between the measured and ERA5-derived RHi distribution and corrects for average biases (Teoh et al., 2022a), we note that it does not correct for the RHi errors at specific waypoints (Teoh et al., 2024a). Thus, RHi uncertainties at each waypoint can remain significant.

### 2.2.3 Contrail cirrus prediction model

Contrails form when the ambient temperature ($T_{\text{amb}}$) at the flight waypoint is below the $T_{\text{SAC}}$ which is estimated by

$$T_{\text{SAC}}[\text{K}] = (273.15 - 46.46) + 9.43\ln(G - 0.053) + 0.72[\ln(G - 0.053)]^2, \tag{2}$$

where $G$ is the gradient of the mixing line in a temperature-humidity diagram,

$$G = \frac{\text{EI}_{\text{H}_2\text{O}} \, p_{\text{amb}} \, c_{\text{p}} \, R_1}{Q_{\text{fuel}} \, (1-\eta) \, R_0}. \tag{3}$$

$\text{EI}_{\text{H}_2\text{O}}$ is the water vapour emissions index and assumed to be 1.237 kg kg$^{-1}$ for Jet A-1 (Gierens et al., 2016), $\eta$ is provided by the aircraft performance model (Section 2.2.1), $p_{\text{amb}}$ is the pressure altitude at each waypoint, $c_{\text{p}}$ is the isobaric heat capacity of dry air (1004 J kg$^{-1}$ K$^{-1}$), and $R_1$ (461.51 J kg$^{-1}$ K$^{-1}$) and $R_0$ (287.05 J kg$^{-1}$ K$^{-1}$) are the gas constant for water vapour and dry air respectively.

Two successive waypoints that satisfy the SAC forms a contrail segment that can either be short-lived or persistent (Schumann, 1996). A parametric wake vortex model is then used to simulate the wake vortex downwash (Holzapfel, 2003), of which CoCiP assumes that the process is instantaneous and does not resolve the temporal evolution of the wake vortex (Schumann, 2012).

Persistent contrails in CoCiP are defined when their post-wake vortex ice water content (IWC) remains above $10^{-12}$ kg kg$^{-1}$. The persistent contrail width ($W$) and depth ($D$) in CoCiP, defined as the dimensions along the y- and z-axis of a Gaussian plume, are initialised as,

$$W_{t=0} = \frac{\pi}{4} S_a,$$ (4)

$$D_{t=0} = 0.5 \times dZ_{max},$$ (5)

where $S_a$ is the aircraft wingspan and $dZ_{max}$ is the maximum vertical displacement of the contrail mid-point after the wake vortex breakup.

The evolution of different contrail properties is then simulated using a first order Euler method with model time steps ($dt$) of 40 s. More specifically, the change in contrail dimensions over time are estimated as,

$$W_t = \sqrt{8\sigma_{yy}},$$ (6)

$$D_t = \sqrt{8\sigma_{zz}},$$ (7)

where σ is a dispersion matrix that captures the spread of the contrail plume along the y- and z-axes. σ is influenced by various factors such as wind shear, contrail segment length, diffusivity, and $dt$ (Schumann, 2012). CoCiP assumes that the contrail segment is sublimated when its ice particle number concentration or optical depth drops below $10^3$ m$^{-3}$ and $10^{-6}$, respectively, or when the mid-point of the contrail plume advects beyond the simulation domain (40 – 60° N and 10° W – 10° E). We specifically selected a $dt$ that is significantly smaller than the typical range that was used in previous studies (1800–3600 s) (Schumann et al., 2015; Teoh et al., 2020a, 2022a) to superimpose the simulated contrail outputs to the video footage and perform a more comprehensive assessment of the early-stage contrail evolution.

## 2.3 Camera transformation model

Before comparing the camera observations with aircraft positions and simulated CoCiP outputs, we first correct any radial and tangential distortion of the video footage using the OpenCV homography method (Bradski, 2000), specifically applying the chessboard calibration technique (Tsai, 1987; Wu et al., 2015) described in Appendix A2. After correcting for distortions, we project the ADS-B waypoints and simulated contrail dimensions onto the video footage using a camera transformation model that follows a two-step process. First, the real-world 3D positions (i.e., ADS-B waypoints and the simulated mid-point and edges of the contrail plumes) are mapped into a 3D camera coordinate system ($X, Y, Z$) using an extrinsic (rotation) matrix. Next, the 3D camera coordinates ($X, Y, Z$) are transformed into a 2D pixel coordinate system ($u, v$) using an intrinsic (camera) matrix. Using this two-step process, Fig. 2 shows the ADS-B waypoints and simulated contrails superimposed onto the video footage, specifically young contrails less than 6 minutes old that were formed within the camera's field of view. Similarly, Fig. 3 projects the simulated dimensions of aged contrails (i.e., those initially formed outside the camera's field of view and

subsequently advected into it) onto the footage and compares them with the observed contrails. Further details of the camera transformation model can be found in Appendix A3.

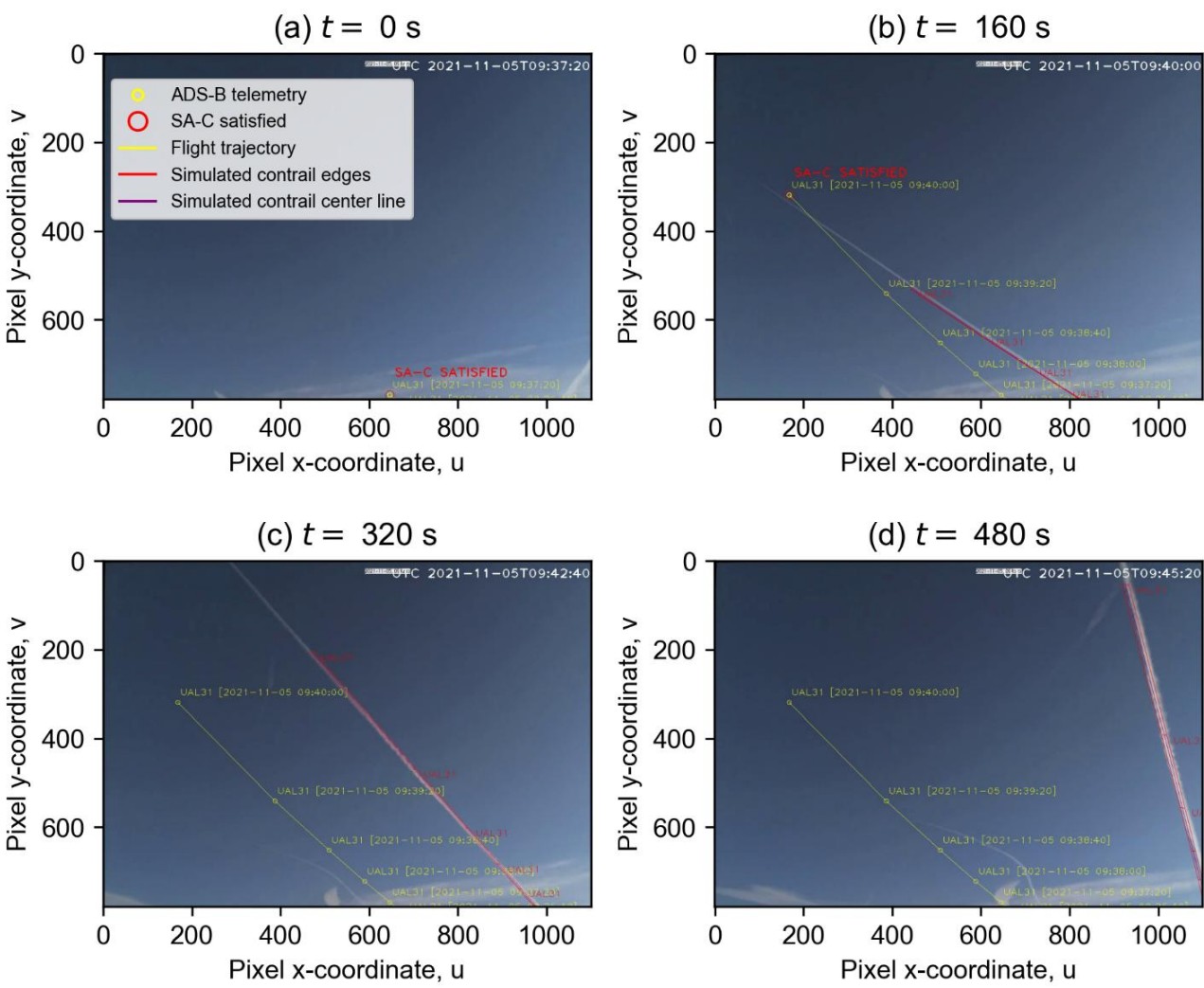

**Figure 2: Example of the flight trajectory and simulated contrail dimensions from the flight with callsign "UAL31", both of which are superimposed onto the video footage using the camera transformation model (detailed in Section 2.3). The flight trajectory and persistent contrails were observed on 5-Nov-2021 between 09:37:20 and 09:45:20 (UTC).**

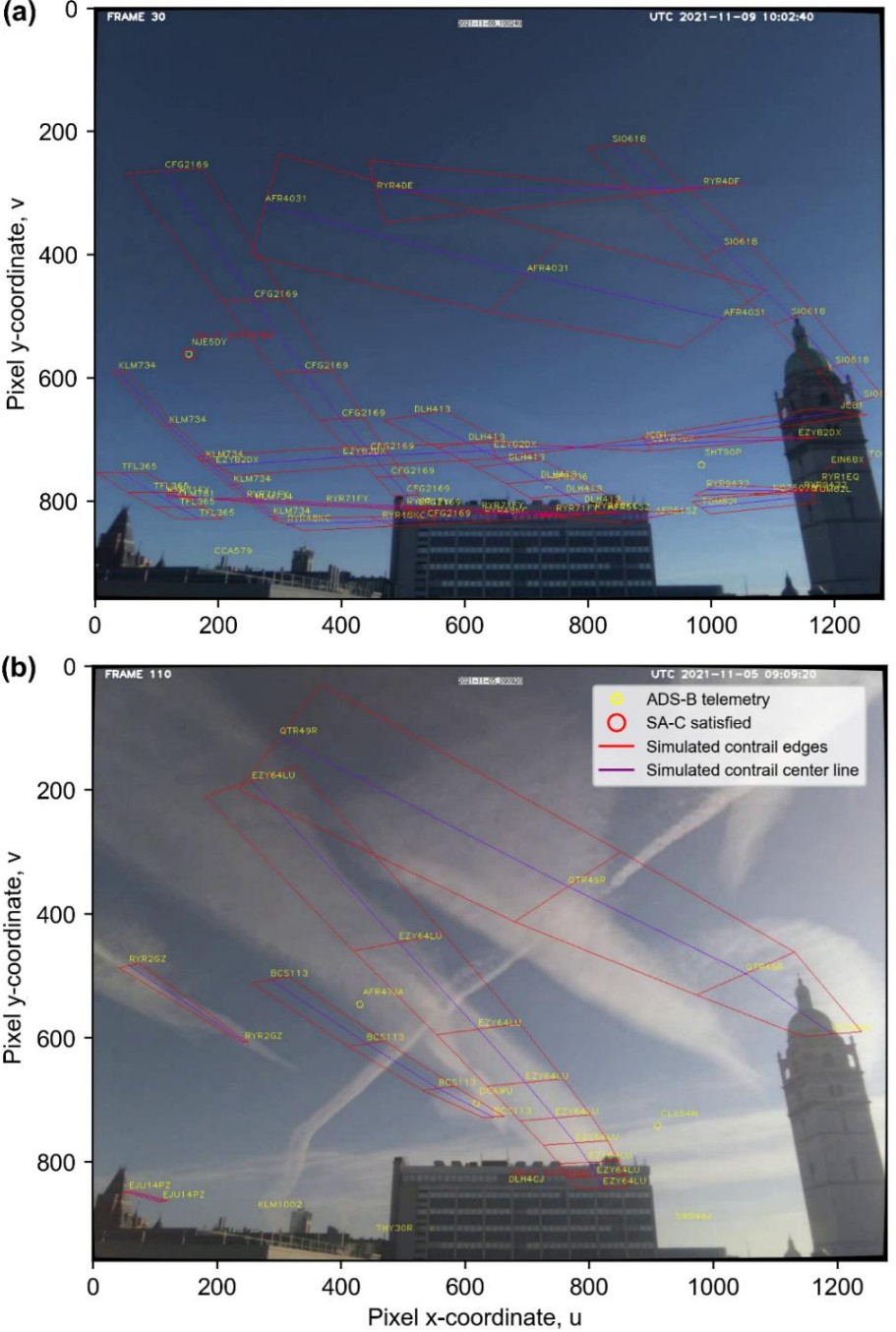

**Figure 3: Examples of the simulated contrails that were initially formed outside the camera's observation domain and subsequently drifted into view on: (a) 9-Nov-2021 at 10:02:40 UTC; and (b) 5-Nov-2021 at 09:09:20 UTC. The CoCiP-simulated contrail dimensions are superimposed onto the video footage using the camera transformation model (detailed in Section 2.3). In panel (a), the faint signals and absence of observed contrails suggest that they could be false positive outcomes ($N_{Camera}$ & $Y_{Sim=CoCiP}$). In panel (b), the absence of labels on some observed contrails indicates that they were most likely false negative outcomes ($Y_{Camera}$ & $N_{Sim=CoCiP}$).**

## 2.4 Comparison between contrail observation and simulation

We visually compare the simulated contrail formation with observations and classify each waypoint into four groups: (i) true positive cases ($Y_{Camera}$ & $Y_{Sim}$), where contrails are both observed by the camera ($Y_{Camera}$) and predicted in the simulation ($Y_{Sim}$); (ii) true negative cases ($N_{Camera}$ & $N_{Sim}$), where no contrails are observed ($N_{Camera}$) and predicted ($N_{Sim}$); (iii) false positive cases ($N_{Camera}$ & $Y_{Sim}$), where contrails are predicted in the simulation but not observed; and (iv) false negative cases ($Y_{Camera}$ & $N_{Sim}$), where contrails are observed but not predicted in the simulation. More specifically, we evaluate the accuracy of the contrail simulation workflow by first assessing whether it correctly identifies short-lived contrails based on the SAC (i.e., $T_{amb} < T_{SAC}$), noting correct and incorrect predictions as $Y_{Sim=SAC}$ and $N_{Sim=SAC}$, respectively. Additionally, we also compare CoCiP's definition of persistent contrail formation (i.e., post wake vortex contrail IWC $> 10^{-12}$ kg kg$^{-1}$) against observations, with accurate and missed predictions denoted as $Y_{Sim=CoCiP}$ or $N_{Sim=CoCiP}$, respectively. In instances where multiple observed contrail segments ($Y_{Camera}$) overlap and/or are closely clustered together, we assign them to the respective ADS-B waypoints through manual visual inspection of preceding frames (Segrin et al., 2007).

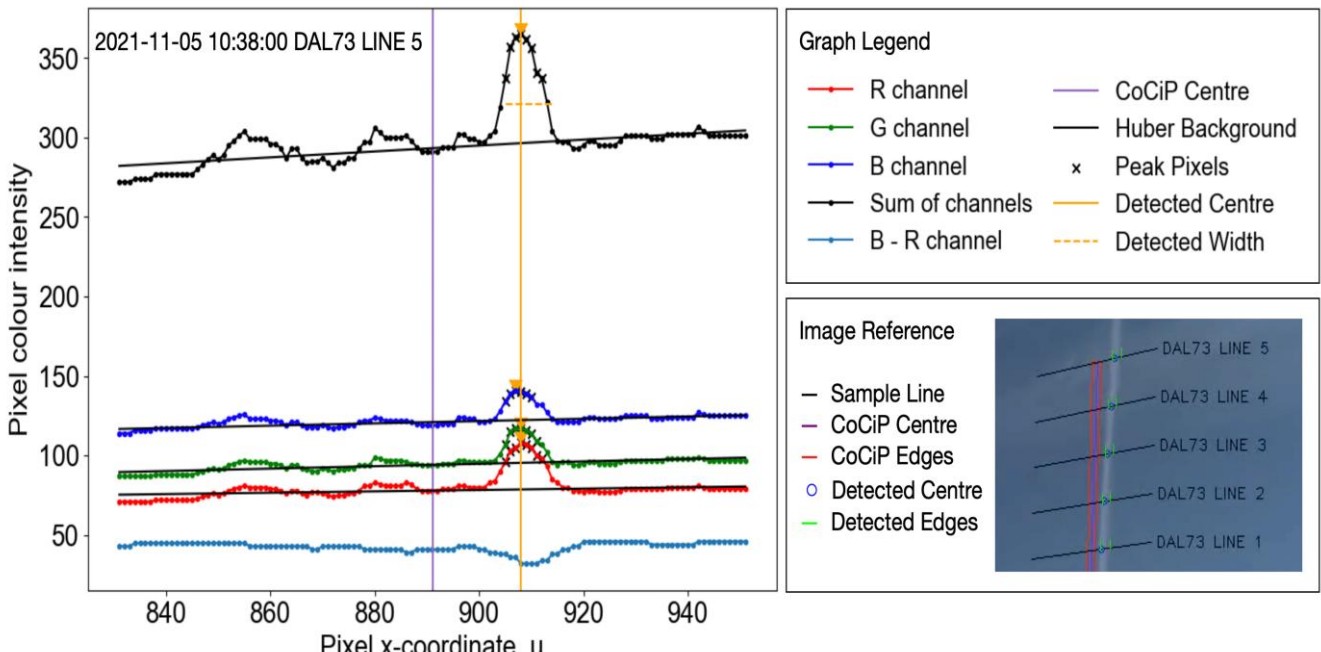

**Figure 4: Pixel colour intensity profiles of the contrail waypoint at Line 5 (shown at the bottom right panel). The contrail observed in the bottom right panel was formed by the flight with callsign "DAL73". In the left panel, the black linear trendlines represent the best-fit background colour intensity for each RGB channel. The solid yellow vertical line marks the mid-point of the observed contrail plume, while the dashed (horizontal) yellow line indicates the estimated contrail pixel width. In both the left and bottom right panels, the purple line indicates the centre of the simulated contrail plume from CoCiP, and in the bottom right panel, the red lines show the simulated contrail edges.**

All waypoints with $Y_{Camera}$ are further classified into three categories based on their observed contrail lifetime defined as the duration during which the contrail is observed by the camera: (i) short-lived contrails with lifetimes of fewer than 2 minutes;

(ii) contrails with lifetimes of between 2 and 10 minutes; and (iii) persistent contrails with lifetimes of least 10 minutes (World Meteorological Organization, 2017). We note that the observed contrail lifetime in our study is restricted by the contrail either advecting out of the camera's field of view (see Fig. A2), becoming too small or faint to be visible in the footage, or sublimating within the observation domain.

Additionally, for waypoints with true positive cases ($Y_{Camera}$ & $Y_{Sim=CoCiP}$), we also compare their observed lifetimes and evolving contrail width relative to the simulated CoCiP outputs. To estimate the observed contrail pixel width from the video footage, we apply the Bresenham (2010) line drawing algorithm at each ADS-B waypoint to extract: (i) a line of pixels orthogonal to the flight trajectory; and (ii) the Red-Green-Blue (RGB) colour channel intensity of these pixels (Fig. 4). Previous studies found that the presence of clouds can be identified by their prominent increase in pixel intensity, especially in the red

channel relative to the blue channel, because the sky scatters more blue than red light while clouds scatter both red and blue light equally (Long et al., 2006; Shields et al., 2013). However, due to day-to-day variability in atmospheric conditions, we were unable to consistently identify contrails from the video footage by applying a fixed threshold for the red-blue pixel intensity ratio. Instead, we compare the relative difference between the local pixel intensity ($P_{u,v}$) and the estimated background pixel intensity ($\hat{P}_{u,v}^{B}$), i.e., the estimated pixel intensity of the background sky assuming that the contrail is absent,

$$\Delta P_{u,v} = P_{u,v} - \hat{P}_{u,v}^{B}. \tag{8}$$

Here, $\hat{P}_{u,v}^{B}$, represented by the black line of best fit in the RGB plot of Fig. 4, is estimated using a Huber regression instead of a traditional least squares regression to minimise the regression sensitivity to outliers (Pedregosa et al., 2012). The observed contrail pixel width at each waypoint and time slice is then estimated from the video footage as follows,

$$\Delta P_{u,v} > \overline{\Delta P_{u,v}} + 2\sigma(\Delta P_{u,v}), \tag{9}$$

where $\overline{\Delta P_{u,v}}$ and $\sigma(\Delta P_{u,v})$ are the mean and standard deviations of the line of pixels orthogonal to the flight trajectory respectively, and the mid-point of the observed plume determined by locating the local maximum of $\Delta P_{u,v}$ (Fig. 4). The reverse camera transformation is then applied to convert the 2D plane pixel width to a geometric width within a 3D space. Notably, due to the lack of depth information from a single camera, we assume that observed contrail altitude is equal to the modelled contrail altitude from CoCiP. This assumption introduces an additional source of error in the observed geometric contrail width

when compared to the pixel contrail width, which we discuss in Section 3.3.

## 3 Results and discussion

Section 3.1 compares the observed contrail formation with those predicted by the SAC and CoCiP. Section 3.2 evaluates the observed contrail lifetime against the ERA5-derived meteorology and simulated contrail lifetime, while Section 3.3 compares the temporal evolution of contrail width between the observation and simulation. Finally, Section 3.4 briefly explores the

potential limitations in detecting contrails from the video footage. Across these sections, we discuss the known and potential

factors that may contribute to the discrepancies between the observed and simulated contrail properties, while acknowledging that the list of factors may not be exhaustive.

## 3.1 Contrail formation

A total of 1,582 unique waypoints from 281 flights were identified across five days of video footage. Contrail formation was
observed in 59.6% of these waypoints ($Y_{Camera}$), 81.6% of these waypoints satisfied the SAC in the simulation ($Y_{Sim=SAC}$), and 44.2% formed persistent contrails according to CoCiP's definition ($Y_{Sim=CoCiP}$) (Table 1).

**Table 1: Summary statistics for each day when contrails were observed by the camera. For each of the five days, the observed contrail formation from the video footage is compared with the two different definitions of contrail formation in the simulation, i.e., using the SAC ($T_{amb} < T_{SAC}$) and CoCiP's definition of persistent contrail formation (post wake vortex contrail IWC > $10^{-12}$ kg kg$^{-1}$).**

| Date | 05-Nov-2021 | 09-Nov-2021 | 14-Jan-2022 | 26-Feb-2022 | 10-Apr-2022 | TOTAL |
|---|---|---|---|---|---|---|
| Times (UTC) | 09:00 – 11:00 | 09:00 – 11:00 | 10:00 – 14:00 | 07:00 – 09:00, 11:00 – 12:00 | 08:00 – 11:00 | - |
| Hours | 2 | 2 | 4 | 3 | 3 | **14** |
| Number of flights | 62 | 39 | 38 | 73 | 69 | **281** |
| Number of waypoints | 317 | 223 | 210 | 419 | 413 | **1582** |
| d$T_{SAC}$, all waypoints (K)[a] | -3.0 ± 7.3 | -7.5 ± 8.7 | -3.2 ± 10.9 | -8.6 ± 11.5 | -6.3 ± 10.3 | **-6.0 ± 10.2** |
| RHi, all waypoints[a] | 0.80 ± 0.56 | 0.85 ± 0.22 | 0.61 ± 0.15 | 0.61 ± 0.17 | 1.0 ± 0.26 | **0.78 ± 0.35** |
| | | | Contrail formation[b] | | | |
| P($Y_{Camera}$ & $Y_{Sim=SAC}$) | 38.9% | 62.8% | 57.1% | 61.6% | 68.8% | **58.5%** |
| P($N_{Camera}$ & $Y_{Sim=SAC}$) | 45.3% | 17.5% | 16.2% | 18.9% | 16.9% | **23.1%** |
| P($Y_{Camera}$ & $N_{Sim=SAC}$) | 0.9% | 0.4% | 0.0% | 3.1% | 0.0% | **1.1%** |
| P($N_{Camera}$ & $N_{Sim=SAC}$) | 14.9% | 19.3% | 26.7% | 16.4% | 14.3% | **17.3%** |
| Correct prediction[d] | 53.8% | 82.1% | 83.8% | 78.0% | 83.1% | **75.8%** |
| | | | Contrail persistence[c] | | | |
| P($Y_{Camera}$ & $Y_{Sim=CoCiP}$) | 26.9% | 53.4% | 0.0% | 44.9% | 52.1% | **38.4%** |
| P($N_{Camera}$ & $Y_{Sim=CoCiP}$) | 7.6% | 10.7% | 0.0% | 0.7% | 9.7% | **5.7%** |
| P($Y_{Camera}$ & $N_{Sim=CoCiP}$) | 13.0% | 9.9% | 57.1% | 19.8% | 16.7% | **21.2%** |
| P($N_{Camera}$ & $N_{Sim=CoCiP}$) | 52.5% | 26.0% | 42.9% | 34.6% | 21.5% | **34.7%** |
| Correct prediction[d] | 79.4% | 79.4% | 42.9% | 79.5% | 73.6% | **73.1%** |

[a]: Mean and one standard deviation across all waypoints, as derived from the ERA5 HRES. For each of the five days, the ambient meteorological conditions across all flight waypoints are visualised in Fig. 6.

[b]: Contrail formation in the simulation is determined by the SAC, where $Y_{Sim=SAC}$ denotes that $T_{amb} < T_{SAC}$, and $N_{Sim=SAC}$ denotes that $T_{amb} \geq T_{SAC}$.

[c]: Contrail persistence in the simulation is determined by CoCiP, where $Y_{Sim=CoCiP}$ denotes that the post wake vortex contrail IWC $\geq 10^{-12}$ kg kg$^{-1}$, and $N_{Sim=CoCiP}$ denotes that the contrail IWC $< 10^{-12}$ kg kg$^{-1}$.

[d]: The correct prediction is calculated by ($Y_{Camera}$ & $Y_{Sim}$) + ($N_{Camera}$ & $N_{Sim}$)

When evaluated using the SAC, the simulation correctly predicted contrail formation and absence for 75.8% of the waypoints, i.e., true positives ($Y_{Camera}$ & $Y_{Sim=SAC}$ = 58.5%) plus true negatives ($N_{Camera}$ & $N_{Sim=SAC}$ = 17.3%), of which: (i) true positive waypoints are always formed above 30,000 feet; while (ii) true negative waypoints were always formed below 32,000 feet

where warmer temperatures limits contrail formation, or above 40,000 feet where drier stratospheric conditions are more common (Fig. 5a). In contrast, the SAC incorrectly predicted contrail formation in 24.2% of the waypoints, where the false positives ($N_{Camera}$ & $Y_{Sim=SAC}$ = 23.1%) significantly outweigh the false negatives ($Y_{Camera}$ & $N_{Sim=SAC}$ = 1.1%). This overestimation in contrail formation by the SAC may be due to observation challenges, as false positive waypoints were often associated with very low RHi's (0.62 ± 0.38 at 1σ, Fig. 6b) relative to true positive waypoints (0.90 ± 0.30 at 1σ, Fig. 6a), potentially resulting in very short-lived or faint contrails that might not be detected by cameras (Fig. 3a). Other factors that may influence the SAC accuracy include uncertainties in: (i) $T_{amb}$ from the ERA5 HRES; (ii) $T_{SAC}$, resulting from modelling errors in η, c.f. Eq. (2) and (3), and the assumption of homogenous plume mixing; and (iii) soot activation at $T_{amb} \approx T_{SAC}$, which are likely incomplete (Bräuer et al., 2021) and becomes strongly dependent on the soot dry core radius and hygroscopicity that are not accounted for by the SAC (Bier et al., 2022). Indeed, contrails at waypoints with incorrect predictions were generally formed at higher temperatures ($dT_{SAC} = T_{amb} - T_{SAC}$ = -7.8 ± 4.3 K at 1σ) compared to true positive waypoints ($dT_{SAC}$ = -12.8 ± 3.7 K at 1σ) (Fig. 5a).

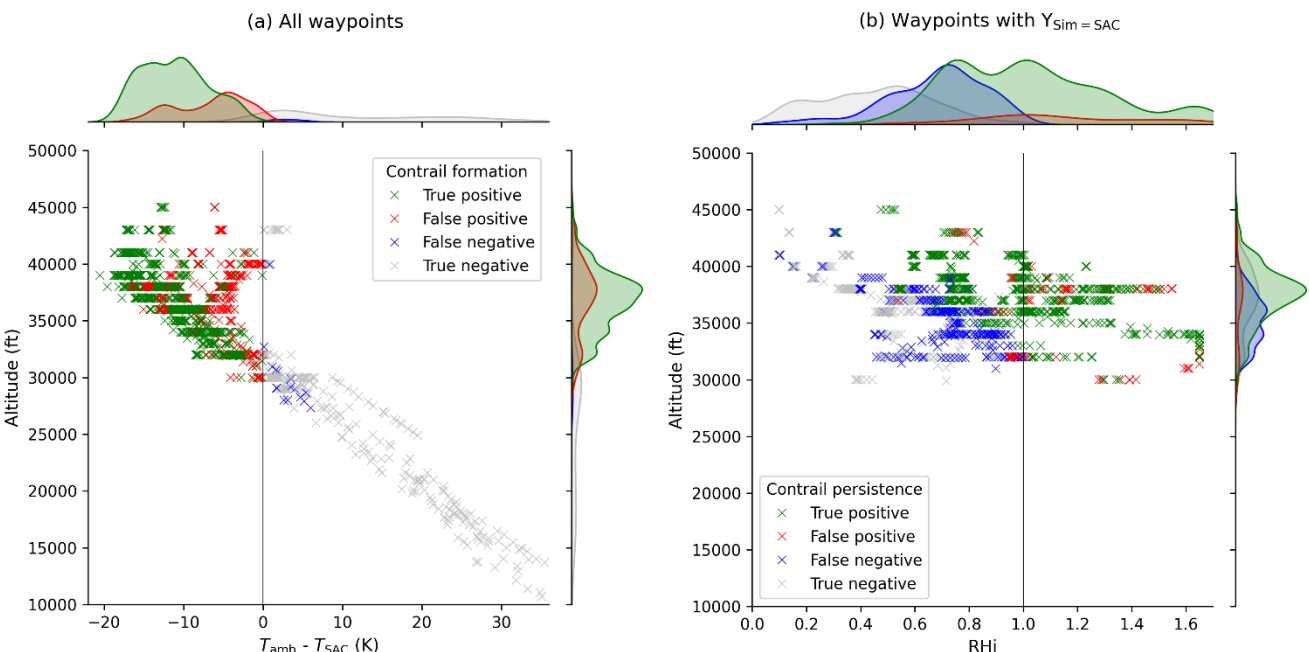

**Figure 5: Joint plot of the aircraft barometric altitude versus the: (a) difference between the ambient ($T_{amb}$) and SAC threshold temperature ($T_{SAC}$) across all flight waypoints; and (b) corrected RHi from the ERA5 HRES for waypoints that satisfy the SAC in the simulation ($Y_{Sim=SAC}$). In both panels, green data points represent true positive outcomes ($Y_{Camera}$ & $Y_{Sim}$), red for false positive outcomes ($N_{Camera}$ & $Y_{Sim}$), blue for false negative outcomes ($Y_{Camera}$ & $N_{Sim}$), and grey for true negative outcomes ($N_{Camera}$ & $N_{Sim}$). In panel (b), the false negative ($Y_{Camera}$ & $N_{Sim=CoCiP}$) and true negative outcomes ($N_{Camera}$ & $N_{Sim=CoCiP}$) correspond to waypoints that satisfied the SAC in the simulation but did not persist beyond the wake vortex phase.**

CoCiP defines persistent contrail formation as occurring when the post wake vortex contrail IWC exceeds $10^{-12}$ kg kg$^{-1}$ ($Y_{Sim=CoCiP}$), and adiabatic heating from the wake vortex downwash is assumed to occur instantaneously at the time of contrail

initialisation. As a result, waypoints with $Y_{Sim=CoCiP}$ are a subset of $Y_{Sim=SAC}$. Using CoCiP's definition of persistent contrails, the overall accuracy of contrail predictions over five days decreased slightly from 75.8% (SAC approach) to 73.1%, with significant variability between individual days (Table 1). Unlike with the SAC, the percentage of false negative waypoints ($Y_{Camera}$ & $N_{Sim=CoCiP}$ = 21.2%) is nearly four times higher than the false positive waypoints ($N_{Camera}$ & $Y_{Sim=CoCiP}$ = 5.7%) (c.f.

$Y_{Camera}$ & $N_{Sim=SAC}$ = 1.1% vs. $N_{Camera}$ & $Y_{Sim=SAC}$ = 23.1%). False negative waypoints also tend to occur at lower altitudes (35100 ± 2600 feet at 1σ) and at sub-saturated RHi conditions (0.68 ± 0.19 at 1σ) relative to those with true positive outcomes (37500 ± 2700 feet and 1.02 ± 0.29) (Fig. 5b). Notably, on 14-Jan-2022, correct contrail predictions dropped sharply from 83.8% to 42.9%, with no persistent contrails predicted in the simulation, because the ERA5-derived RHi at all waypoints were well below ice supersaturation (0.07–0.79, Fig. 6).The difference in accuracy between the SAC and CoCiP's definition of

persistent contrail formation is most likely due to contrail model simplifications (i.e., instantaneous wake vortex downwash) which can underestimate the simulated contrail lifetimes, particularly for short-lived contrails. Indeed, when waypoints are segmented by the observed contrail lifetime, the simulation correctly predicted contrail formation for only 55% of waypoints with short-lived contrails ($Y_{Camera}$ < 2 minutes & $Y_{Sim=CoCiP}$). However, correct predictions increased significantly to 96% for waypoints with observed lifetimes between 2 and 10 minutes, and to 86% for waypoints with observed contrails persisting

beyond 10 minutes.

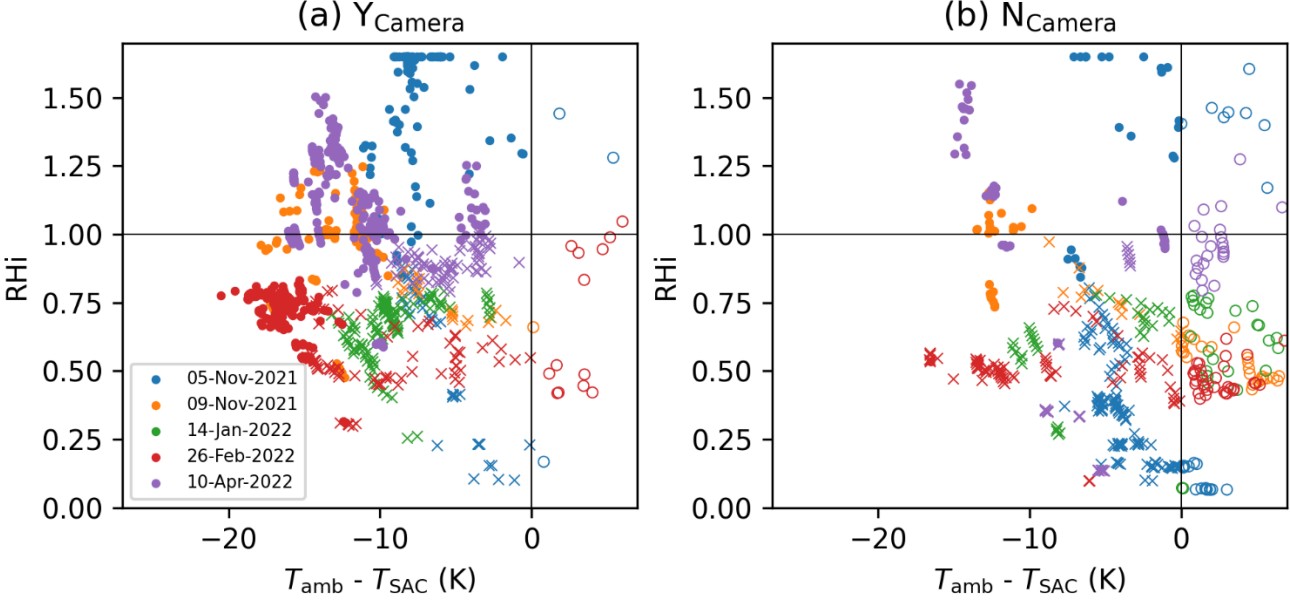

**Figure 6: Corrected RHi from the ERA5 HRES versus the difference between the ambient ($T_{amb}$) and SAC threshold temperature ($T_{SAC}$) for all waypoints across five days: (a) with; and (b) without contrails observed from the video footage. In both plots, data points with no fill (circles) represent waypoints where contrails did not form in the simulation ($N_{Sim=SAC}$), crosses indicate waypoints**

**that satisfied the SAC in the simulation ($Y_{Sim=SAC}$), and filled data points denote waypoints where persistent contrails were formed in the simulation ($Y_{Sim=CoCiP}$).**

## 3.2 Contrail lifetime

We categorise the 942 unique waypoints with observed contrails ($Y_{\text{Camera}}$) into three groups based on their observed contrail lifetimes (Section 2.4). Among these waypoints, 73.3% of them are short-lived with observed lifetimes of less than 2 minutes.

Of these short-lived contrails, 99.3% of them either became too small to be tracked or sublimated within the camera's field of view, while 0.7% advected out of it. Contrails with observed lifetimes ranging between 2 and 10 minutes made up 12.5% of the observations, with 36% of them drifting beyond the camera's field of view. The remaining 14.2% of contrails had observed lifetimes exceeding 10 minutes, of which 64% of them advected beyond the camera's field of view.

For waypoints with $Y_{\text{Camera}}$, we compared their observed contrail lifetimes against the ERA5-derived meteorology at the point and time of their formation (Fig. 7). Our analysis shows that: (i) 98% of the observed contrails fulfilled the SAC ($T_{\text{amb}} < T_{\text{SAC}}$) in the simulation; (ii) 78% of short-lived contrails ($Y_{\text{Camera}} < 2$ minutes) were formed under ice sub-saturated conditions (RHi $< 100\%$), with a mean RHi of $81 \pm 25\%$ ($1\sigma$); (iii) 59% of contrails with observed lifetimes of between 2 and 10 minutes also formed under ice sub-saturated conditions, but the mean RHi is higher at $103 \pm 32\%$; and (iv) 75% of persistent contrails ($Y_{\text{Camera}} > 10$ minutes) were formed in ice supersaturated conditions (RHi $> 100\%$), with a mean RHi of $124 \pm 26\%$.

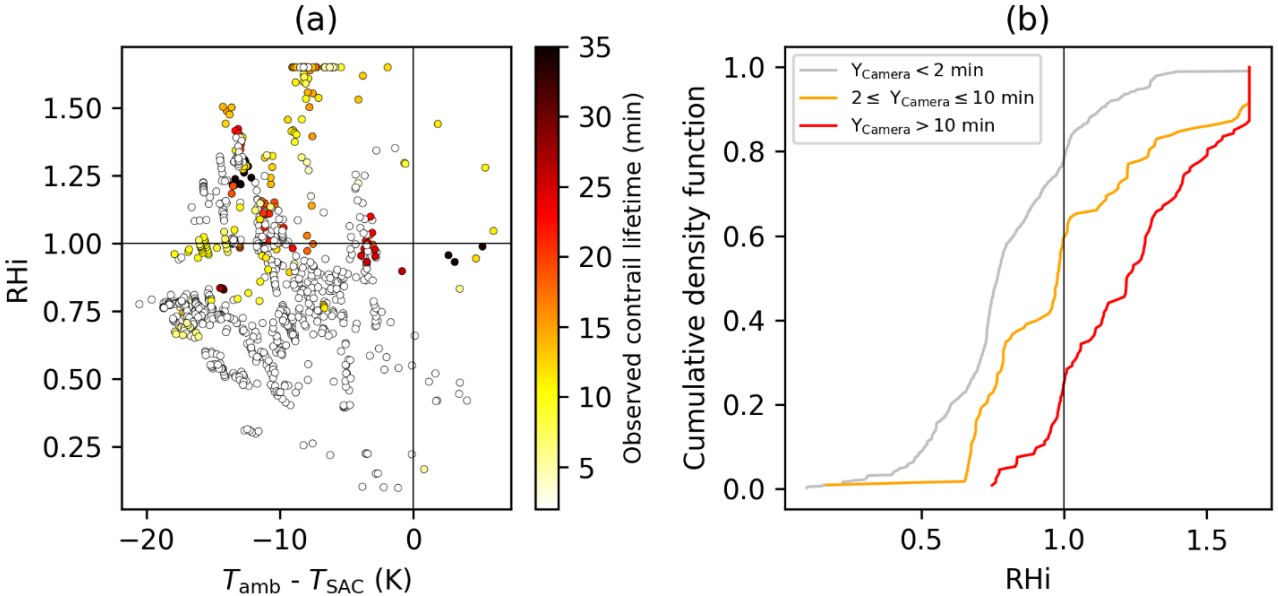

**Figure 7: Evaluation of the observed contrail lifetime relative to the ERA5-derived meteorology at the point and time of their formation for all waypoints with observed contrails ($Y_{\text{Camera}}$). Panel (a) compares the observed contrail lifetime with the RHi (y-axis) and the difference between the ambient temperature ($T_{\text{amb}}$) and SAC threshold temperature ($T_{\text{SAC}}$) (x-axis). Panel (b) shows the cumulative density functions of the initial RHi, with the data points segmented into three groups based on their observed contrail lifetimes, i.e., those lasting fewer than 2 minutes (gray), between 2 and 10 minutes (orange), and more than 10 minutes (red).**

Fig. 8 shows a poor visual agreement between the observed and simulated contrail lifetime, with the simulated lifetimes being strongly influenced by the ERA5-derived RHi. Specifically, the simulation always predicts contrails with lifetimes below 5 minutes when the RHi is less than 100%, often underestimating the observed contrail lifetimes. Additionally, the simulation

consistently predicts contrails with lifetimes exceeding 2 minutes when the RHi is above 100%, even though around half of these waypoints were observed with short-lived contrails (< 2 minutes). It also tends to predict contrail lifetimes longer than 35 minutes when the RHi exceeds 120%, though evaluating these predictions is challenging as the maximum observed contrail lifetime can be limited by the contrail drifting out of the field of view or becoming too small or faint to be tracked (Fig. 3a). Two known factors contribute to the uncertainty in ERA5-derived RHi estimates. Firstly, the ERA5 HRES humidity fields often produce weakly supersaturated RHi estimates (Agarwal et al., 2022; Reutter et al., 2020; Teoh et al., 2022a). Although corrections were applied to ensure that the ERA5-derived RHi distribution is consistent with in-situ measurements (Section 2.2.2), RHi uncertainties remain large at the waypoint level (Teoh et al., 2024a). Secondly, the spatial resolution of the ERA5 HRES (0.25° longitude × 0.25° latitude ≈ 18 × 28 km) is insufficient to capture the sub-grid scale RHi variabilities that have been observed from in-situ measurements (Wolf et al., 2024). Given the small study domain, where the camera's field of view fits within 10 grid boxes of the ERA5 HRES (Fig. A2), our simulation would be particularly impacted by these sub-grid scale effects. However, we do not evaluate these effects due to our small sample size (n = 942 for waypoints with $Y_{Camera}$ distributed over 14 h and 10 grid boxes). The observed decline in agreement between observations and simulations from contrail formation (Table 1) to contrail persistence (Fig. 8) is consistent with earlier studies that found the ERA5 HRES temperature fields to be more accurate than its humidity fields (Gierens et al., 2020; Reutter et al., 2020).

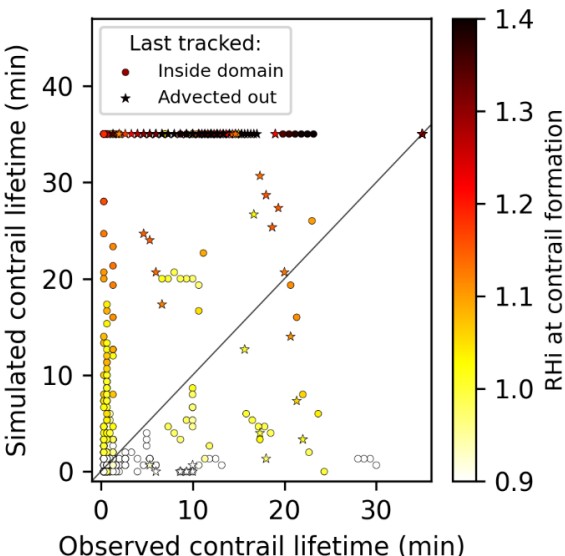

**Figure 8: Comparison between the observed and simulated contrail lifetime for waypoints with true positive outcomes ($Y_{Camera}$ & $Y_{Sim=CoCiP}$). Observed contrails are categorised based on their final known position: circles represent contrails that either sublimated or became too small or faint within the observation domain; while stars indicate that the contrail drifted out of the observation domain and can no longer be tracked. The colour bar represents the corrected ERA5-derived RHi at the time of contrail formation. The simulated contrail lifetime in this plot is constrained to 35 minutes to align with the maximum observed contrail lifetime.**

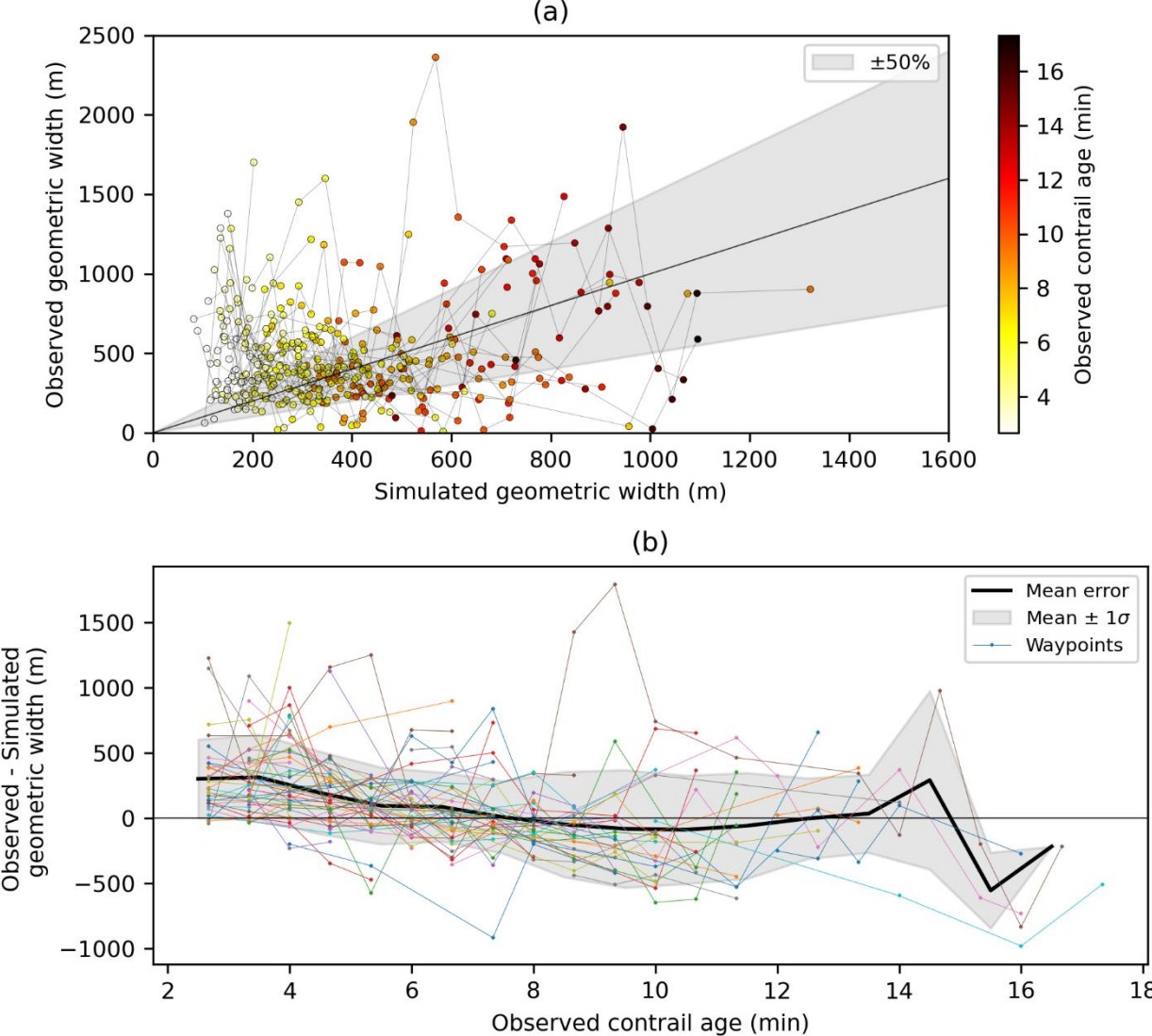

**Figure 9: Comparison between the observed and simulated contrail geometric width for waypoints with true positive cases and with observed lifetimes exceeding 2 minutes ($Y_{Camera}$ > 2 minutes & $Y_{Sim=CoCiP}$). Panel (a) shows a parity plot between the observed and simulated widths at single point in time, with the black line representing the 1:1 line. Panel (b) illustrates the difference between the observed and simulated geometric widths as a function of the observed contrail age. For panels (a) and (b), individual lines connecting different data points represents the temporal evolution of the contrail width at each contrail waypoint. The observed contrail pixel width is converted to the observed geometric width using the reverse camera transformation model (see Section 2.3).**

### 3.3 Contrail width

Figure 9 compares the temporal evolution of the observed and simulated contrail geometric widths for 70 waypoints with true positive cases ($Y_{Camera}$ & $Y_{Sim=CoCiP}$) and observed lifetimes greater than 2 minutes. On average, the simulated contrail geometric widths are around 100 m smaller than the observed widths over the observed contrail lifetime, with the largest

underestimations occurring within the first five minutes (-280 m, on average, Fig. 9b). The tendency to underestimate the simulated contrail widths is consistent with Schumann et al. (2013) and can be attributed to several known factors, including: (i) uncertainties in wind shear and turbulent mixing, where their sub-grid scale variabilities cannot be resolved from the spatiotemporal resolution of the ERA5 HRES (Hoffmann et al., 2019; Paugam et al., 2010; Schumann et al., 2013); (ii) contrail

model errors resulting from the use of simplified physics, such as the Gaussian plume assumption which may not adequately represent the contrail cross-sectional area (Jensen et al., 1998b; Sussmann and Gierens, 1999; Unterstrasser and Gierens, 2010), instantaneous wake vortex assumption, and the initialisation of persistent contrail width solely based on the aircraft wingspan, c.f. Eq. (4), without considering wake vortex dynamics and ambient meteorology (Lewellen and Lewellen, 2001; Schumann, 2012); and (iii) CoCiP's definition of the simulated contrail width (i.e., the length across the y-axis of a Gaussian plume),

which is inherently shorter than the maximum possible observed contrail width (i.e., length across the major axis of an inclined ellipse). These factors are among those identified and may not be exhaustive.

In addition to errors in the simulated contrail width, independent error sources in the observed contrail widths also contribute to the poor visual agreement between the observed and simulated contrail widths (Fig. 9a). Firstly, the presence of other contrails and natural cirrus can affect the Huber regression used to identify the contrail edges, c.f., Eq. (9), thereby contributing

to errors in the observed contrail pixel width (Fig. 3b). Secondly, converting the observed pixel width to geometric width introduces additional errors due to the lack of data on the: (i) contrail altitude, which we assume that the observed contrail altitude is equal to the simulated contrail altitude in CoCiP (Section 2.4); and (ii) inclination angle of the elliptical contrail plume, where parallax errors can contribute to a larger variability in the observed geometric width relative to the pixel width. Assumption (i) is subject to uncertainties in the actual aircraft mass and local meteorology, which can result in additional errors

when simulating the contrail vertical displacement caused by the wake vortex downwash. We evaluate the sensitivity of the observed geometric width to these factors by varying the assumed contrail altitude and the altitude at one of the contrail edges by $\pm$ 100 m. Our results indicate that the inclination angle has a significantly greater influence on the observed contrail geometric width ($\pm$ 36%) compared to the altitude assumption ($\pm$ 0.9%).

### 3.4 Contrail detection limits

We visually examined contrails that initially formed outside the observation domain and were subsequently advected into view, where the results yielded mixed outcomes. Firstly, on 5-Nov-2021 at 09:09:20 UTC, some predicted contrails aligned well with the observations (Fig. 3b). However, not all observed contrails were predicted by the model, and there were notable differences in the locations of predicted and observed contrails. We note that contrail-contrail and cloud-contrail overlapping further complicated the identification of contrail edges and the extraction of contrail widths.

Secondly, on 9-Nov-2021 at 10:02:40 UTC, we were unable to visually confirm the presence of contrails in the video footage (Fig. 3a), despite the simulation predicting contrail cirrus with a mean optical depth of 0.024 [0.002, 0.056] ($5^{th}$ and $95^{th}$ percentile). This suggests that these contrails could be misclassified as false positive cases ($N_{Camera}$ & $Y_{Sim=CoCiP}$) because their optical depths were below or close to the lower visibility threshold limit for ground-based observers (optical depth of $< 0.02$)

(Kärcher et al., 2009). Although faint white grains were visible in the video footage (Fig. 3a), it remains challenging to determine whether these features represent contrail cirrus. This difficulty underscores the challenges that remote sensing methods, including ground-based cameras, have with detecting optically thin contrails below a yet-to-be determined threshold optical depth (Driver et al., 2024; Mannstein et al., 2010; Meijer et al., 2022).

## 4 Discussion and conclusions

Ground-based cameras can observe contrails at a higher spatiotemporal resolution than satellite imagery, making them potentially valuable for validating the early contrail lifecycle as simulated by contrail models. In this study, we develop a methodology to analyse contrail formation, persistence, and their geometric widths from ground-based video footage, and subsequently compare these observations with contrail simulations. Our contrail observations consist of 14 h of video footage recorded on five different days at Imperial College London's South Kensington Campus. The actual flight trajectories intersecting with the camera's field of view were obtained from ADS-B telemetry, and contrails formed by these flights were simulated with CoCiP using historical meteorology from the ECMWF ERA5 HRES reanalysis.

In total, we identified 1,582 flight waypoints from 281 flights from the video footage, with contrails observed in 60% of these waypoints ($Y_{Camera}$) under clear sky conditions. The simulation correctly predicted contrail formation and absence for 76% of these waypoints when evaluated using the SAC ($T_{amb} < T_{SAC}$), and for 73% of waypoints when evaluated using CoCiP's definition of persistent contrail formation (post wake vortex contrail IWC > $10^{-12}$ kg kg$^{-1}$) (Table 1). Among waypoints with incorrect predictions, the SAC overestimated contrail formation, with 23% of waypoints being false positives ($N_{Camera}$ & $Y_{Sim=SAC}$) compared to 1% false negatives ($Y_{Camera}$ & $N_{Sim=SAC}$). In contrast, CoCiP's definition underestimated contrail formation, with 6% of false positives ($N_{Camera}$ & $Y_{Sim=CoCiP}$) versus 21% of false negatives ($Y_{Camera}$ & $N_{Sim=CoCiP}$). A comparison with reanalysis weather data suggests that incorrect predictions were often associated with warmer temperatures ($dT_{SAC}$ = -7.8 ± 4.3 K at 1σ) and sub-saturated RHi conditions (0.68 ± 0.19 at 1σ) relative to those with true positive outcomes ($dT_{SAC}$ = -12.8 ± 3.7 K and RHi = 1.02 ± 0.29) (Fig. 5).

When waypoints with $Y_{Camera}$ are segmented based on their observed contrail lifetime, the simulation accurately predicted contrail formation for only 55% of short-lived contrails ($Y_{Camera}$ < 2 minutes & $Y_{Sim=CoCiP}$), while correct predictions rose to over 85% for contrails with observed lifetimes exceeding 2 minutes ($Y_{Camera}$ ≥ 2 minutes & $Y_{Sim=CoCiP}$). Notably, among the waypoints with $Y_{Camera}$: (i) 98% of them fulfilled the SAC; (ii) 78% of short-lived contrails (observed lifetimes < 2 minutes) initially formed at RHi < 100%; (iii) 59% of contrails with observed lifetimes ranging between 2 and 10 minutes also formed at RHi < 100%; while (iv) 75% of persistent contrails (observed lifetimes > 10 minutes) formed at RHi > 100% (Fig. 7). The observed contrail geometric widths tend to be larger than the simulated widths by an average of 100 m over their observed lifetime, with the most significant underestimations (around 280 m) occurring during the first five minutes (Fig. 9).

Overall, our results show a gradual decline in agreement between observations and simulations, particularly as contrails
progress from formation to persistence. Discrepancies between the observed and simulated contrail properties stem from
multiple sources, including: (i) uncertainties in the ERA5 HRES humidity fields; (ii) sub-grid scale variabilities that cannot be
captured by the spatiotemporal resolution of existing NWP models; (iii) contrail model assumptions and simplifications; (iv)
uncertainties in the simulated aircraft overall efficiency, which influences $T_{SAC}$; (v) observational challenges (Fig. 3); and (vi)
potentially other unidentified factors. Nevertheless, we acknowledge the potential limitations of our study, including the small
sample size and an inherent bias toward selecting contrails formed under high-pressure systems (i.e., clear sky conditions),
which is estimated to account for only 15% of all contrails in the Northern Hemisphere (Bedka et al., 2013). This selection
bias excludes a significant portion of contrails formed in low-pressure systems associated with storms or overcast weather.
Such discrepancies in synoptic weather conditions could introduce varying error patterns in NWP models, which may
propagate and affect the accuracy of the simulated contrail outputs. Additionally, as we specifically selected days with observed
contrails, our findings should not be interpreted as representative of the overall likelihood of contrail formation.

Future work can build upon our research by: (i) developing a methodology to estimate the contrail optical thickness from
ground-based cameras; (ii) establishing a network of ground-based cameras to observe contrails across a larger set of flights
and over a wider domain, while also mitigating the sensitivity of camera models to contrail altitude; (iii) combining ground-
based (i.e., cameras and lidars) and satellite observations to track the whole contrail lifecycle and beyond cloud free conditions;
(iv) conducting a large-scale comparison between the observed and simulated contrails to establish benchmark datasets, which
can be used to validate and improve the accuracy of contrail models and the humidity fields provided by NWP models; and
(v) integrating ground-based observations with contrail forecasts, thereby reducing the uncertainties in the real-time decision
making processes for flight diversions to minimise the formation of strongly warming contrails.

**Appendix**

**A1 Video classification and camera field of view**

Temporal variabilities in weather conditions influence the suitability of the video footage for contrail observations. To filter the video
footage that can be used to observe, track, and extract the properties of contrails, we visually inspect each hourly recording and
classify them based on the background cloud cover (Table A1) and lighting conditions (Table A2). An example of each classification
is shown in Fig. A1. The 14 h of video footage that were selected for further analysis have: (i) clear sky conditions; and (ii) optimal
lighting with strong color contrast between the (blue) sky and (white) contrails. Following the selection of video footages that are
feasible for contrail analysis, we reduced their frame rate to 40 seconds per frame to match the temporal resolution of the ADS-B
data and CoCiP outputs. Figure A2 shows the camera's position and the spatial distribution of observed contrails within its field of
view. The camera transformation model, as will be described in Appendix A3, was applied to systematically superimpose ADS-B
data and CoCiP outputs onto the video footage.

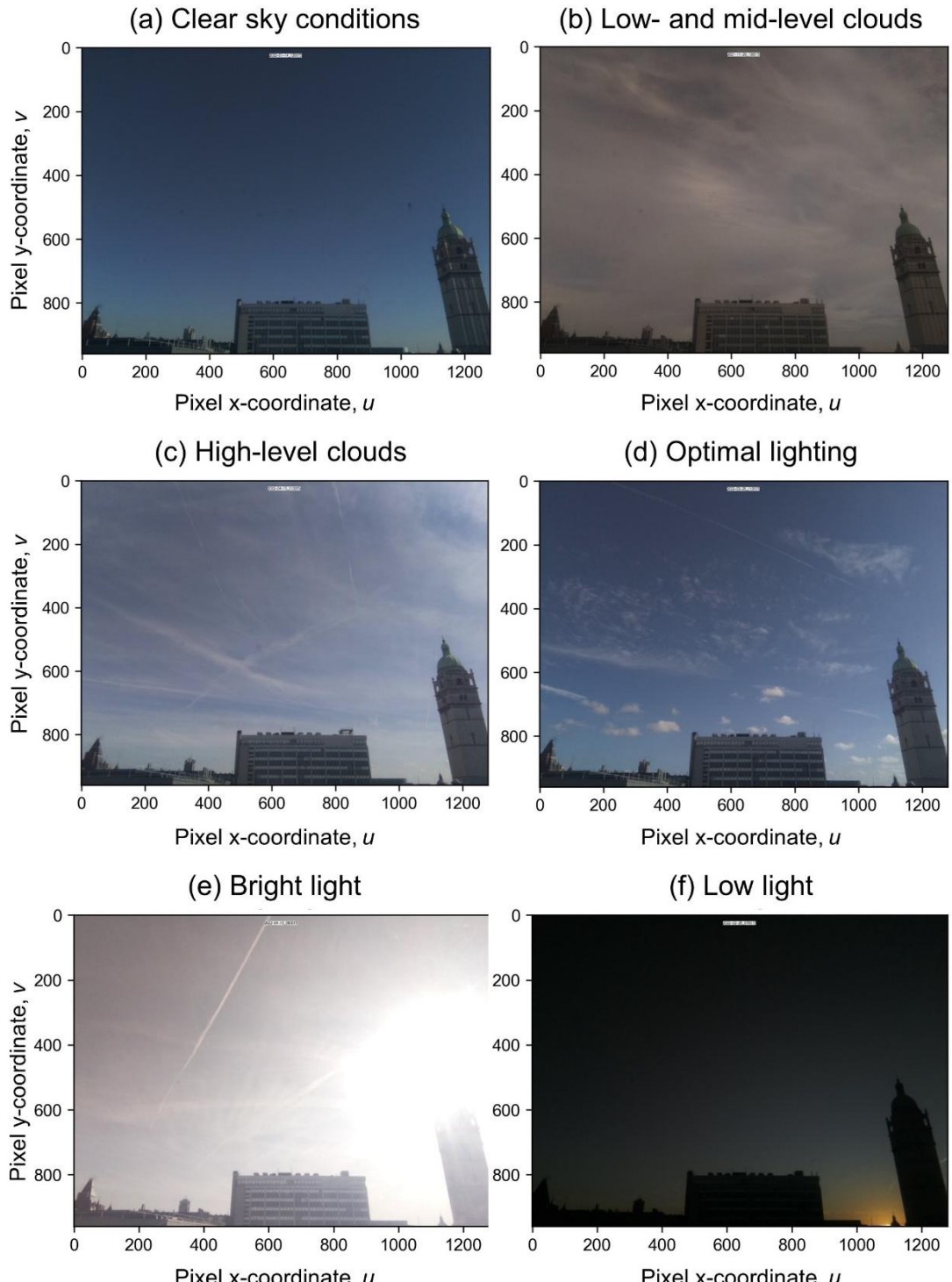

**Figure A1: Examples of the different background cloud cover, i.e., (a) clear sky conditions, (b) low-/mid-level clouds, and (c) high-level clouds), and lighting conditions, i.e., (d) optimal lighting, (e) bright-light; and (f) low-light conditions that were described in Tables A1 and A2.**

**Table A1: Classification of the video footage by the extent of background cloud cover.**

| Category | Remarks/Implications |
|---|---|
| Clear | • Clear sky conditions (0 oktas)* with an absence of low-, mid- and high-level cirrus. |
| Presence of low- and mid-level clouds | • Cloud cover with more than 5 oktas* can potentially obscure contrail observations, thereby limiting the opportunities for analysis. |
| Presence of high-level clouds | • Contrails formed within these clouds may be difficult to identify.<br>• Contrails formed outside and subsequently advected into the camera's field of view may not be easily distinguished from natural cirrus clouds. |

*: The unit "okta" is used to quantify the extent of cloud cover by dividing the sky into eights. A measurement of 0 oktas denotes a completely clear sky, while 8 oktas imply an entirely overcast sky.

**Table A2:** Classification of the video footage by the ambient lighting levels.

| Category | Remarks/Implications |
|---|---|
| Optimal | • Strong color and feature contrast between the (blue) sky and contrails, ideal for contrail observations. |
| Bright light | • Limited color contrast between the (white) sky compared to contrails and natural cirrus clouds.<br>• If the sun is in direct view of the camera, the solar glare may obscure a portion of the image. |
| Low light | • Adjustments to the typical thresholds used to identify contrails will be necessary due to the reduced color brightness of the contrail against a darker background. |

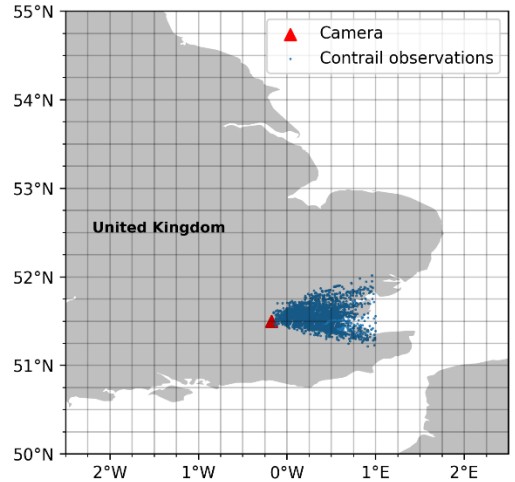

**Figure A2: Location of the camera (51.4988°N, 0.1788°W) and the spatial distribution of observed contrails within its field of view ($n$ = 942 for waypoints with $Y_{Camera}$). The grid boxes represent the spatial resolution of the ERA5 HRES (0.25° longitude × 0.25° latitude). Basemap plotted using Cartopy 0.21.1 © Natural Earth; license: public domain.**

**A2 Corrections to camera distortion**

Unlike the ideal pinhole model, camera images contain radial and tangential distortion. Radial distortion occurs due to the
455 bending of light rays near the edge of a lens, causing straight lines to appear curved. Tangential distortion occurs when lens assembly are not directly parallel and centred over the image plane. Distortion coefficients are determined using a chessboard pattern and homography, and an example process can be found in Wu et al. (2015). Using the OpenCV Python package (Bradski, 2000), every pixel is mapped to a corrected position following these steps:

**STEP 1**: The distorted pixel coordinates ($u_{dist}$, $v_{dist}$) are converted to distorted camera coordinates ($x_{dist}$, $y_{dist}$, $z_{dist}$) in Eq. (A1) using the inverse of the camera intrinsic matrix ($K^{-1}$, see Appendix A3),

$$\begin{bmatrix} x_{dist} \\ y_{dist} \\ z_{dist} \end{bmatrix} = K^{-1} \begin{bmatrix} u_{dist} \\ v_{dist} \\ 1 \end{bmatrix}. \tag{A1}$$

**STEP 2**: The distorted camera coordinates are corrected using Eq. (A2) and Eq. (A3), both of which are found in the OpenCV package documentation,

$$x'' = x'(1 + k_1 r^2 + k_2 r^4 + k_3 r^6) + [2p_1 x'y' + p_2(r^2 + 2x'^2)], \tag{A2}$$

$$y'' = y'(1 + k_1 r^2 + k_2 r^4 + k_3 r^6) + [2p_2 x'y' + p_1(r^2 + 2x'^2)], \tag{A3}$$

where $x' = x_{dist}/z_{dist}$ and $y' = y_{dist}/z_{dist}$ are normalised coordinates, $r = \sqrt{x'^2 + y'^2}$, $k_1 = 0.580$, $k_2 = -2.661$, and $k_3 = 4.420$ are radial distortion coefficients, and $p_1 = 5.803 \times 10^{-1}$ and $p_2 = -2.576 \times 10^{-3}$ are tangential distortion coefficients.

**STEP 3**: The undistorted pixel coordinates ($u$, $v$) are recalculated using Eq. (A4),

$$\lambda \begin{bmatrix} u \\ v \\ z_{dist} \end{bmatrix} = K \begin{bmatrix} x'' \\ y'' \\ z'' \end{bmatrix}. \tag{A4}$$

Figure A3 shows an original frame captured by the camera alongside a corrected frame using the three-step process. While these differences may not be visually discernible, it is crucial to remove distortions to minimise errors when extracting the observed contrail pixel and geometric width from these images. The correction of the minor distortion in the original frame is evident through the added grid lines. All video footage used in the study underwent initial frame-by-frame processing to eliminate distortion before conducting subsequent analysis.

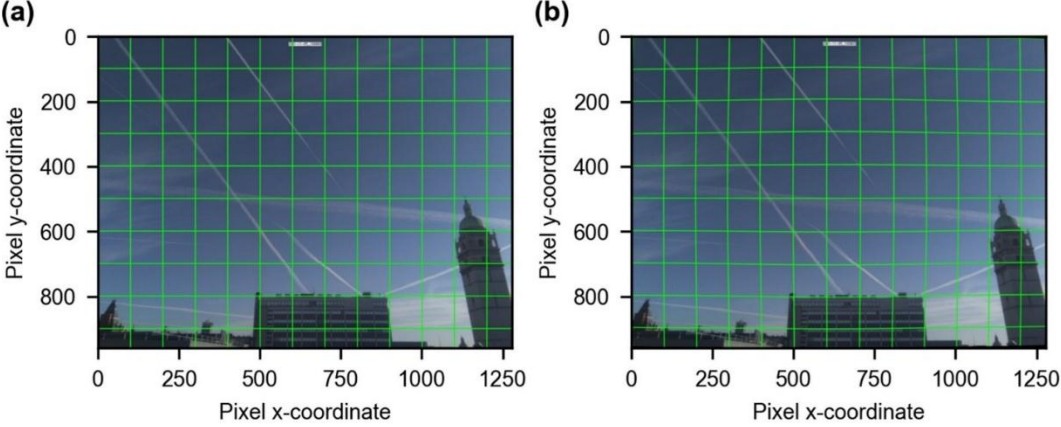

**Figure A3: Side-by-side comparison of (a) an original frame captured by the ground-based camera; and (b) the distortion corrected frame by mapping coordinates to their undistorted positions using the OpenCV Python package.**

## A3 Camera transformation model

After correcting for distortions, a camera transformation method is used to project the aircraft positions and simulated contrail location, which are provided as three-dimensional (3D) positions, to the camera observations which utilises a two-dimensional (2D) pixel coordinate $(u, v)$. A two-step process is used to achieve this:

**STEP 1**: The real-world 3D positions relative to the camera is mapped to a 3D camera coordinate system $(X, Y, Z)$ using an extrinsic (rotation) matrix $R$,

$$R = [R_x][R_y][R_z] = \begin{bmatrix} 0.1434 & -0.1357 & 0.9803 \\ -0.1357 & 0.9785 & 0.1553 \\ -0.9803 & -0.1553 & 0.1219 \end{bmatrix}. \tag{A5}$$

$R$ describes the camera rotation in relation to the world axis, where $R_x$, $R_y$, and $R_z$ are the roll, pitch, and yaw of the camera respectively. The $R$ coefficients are estimated by minimising the residuals between the computed and measured pixel coordinates of known aircraft positions and landmarks that are visible in the camera frame.

**STEP 2**: The 3D camera coordinates is then transformed to a 2D pixel coordinate system $(u, v)$ using an intrinsic (camera)
matrix $K$,

$$K = \begin{bmatrix} f_x & s & x_0 \\ 0 & f_y & y_0 \\ 0 & 0 & 1 \end{bmatrix} = \begin{bmatrix} 708 & 0 & 634 \\ 0 & 708 & 472 \\ 0 & 0 & 1 \end{bmatrix}, \tag{A6}$$

where the camera parameters $f_x$ and $f_y$ are the focal lengths in pixel units, $(x_0, y_0)$ is the principal point of the image, and $s$ represents the axis skew. Fig. 2 in the main text provides an example of the superimposed flight trajectories and simulated contrail properties to the video footage.

## Author contributions

MEJS and EG conceptualised the study. JL, RT, JP, and EG developed the methodology and undertook the investigation. JL, RT, JP, EG, and MS were responsible for software development and data curation. RT and JL created or sourced the figures. RT and JL wrote the original manuscript. MEJS, EG, and MS acquired funding. All authors have read, edited, and reviewed the manuscript, and agreed upon the published version of the paper.

## Funding acknowledgements

EG was supported by a Royal Society University Research Fellowship (URF/R1/191602) and NERC project RECFI-4D (NE/X012255/1). JP was supported by the EPSRC (grant no. EP/S023593/1).

## Data availability

The ADS-B telemetry that is used to derive the actual flight trajectories in this study was purchased from Spire Aviation and can be made available for scientific research upon reasonable request. The pycontrails repository that contains the CoCiP algorithm has recently been published and publicly available at https://doi.org/10.5281/zenodo.7776686. This document used elements of Base of Aircraft Data (BADA) Family 4 Release 4.2 which has been made available by EUROCONTROL to Imperial College London. EUROCONTROL has all relevant rights to BADA. ©2019 The European Organisation for the Safety of Air Navigation (EUROCONTROL). EUROCONTROL shall not be liable for any direct, indirect, incidental, or consequential damages arising out of or in connection with this document, including the use of BADA. This document contains Copernicus Climate Change Service information 2023. Neither the European Commission nor ECMWF is responsible for any use of the Copernicus information. The timelapse videos will be submitted to the Central for Environmental Data Analysis (CEDA) on acceptance.

## Competing interests

There are no conflicts of interest and all funding sources have been acknowledged. All figures are our own. None of the authors has any competing interests.

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
