# Peer review of "Ground-based contrail observations: comparisons with reanalysis weather and contrail model simulations"

_EGUsphere, 2024_

## Referee Comment (RC1)

Review of

**Ground-based contrail observations: comparisons with flight telemetry and contrail model estimates**

by **Low et al.** submitted to **AMT**

**General comments**

The paper reports on the installation of a ground-based camera to take pictures of contrails in the upper troposphere and on the derivation of contrail-relevant statistics and properties from such data. Furthermore, the observations are compared on a statistical basis with model estimates obtained from a simplified contrail plume model.

I commend the authors for their efforts in setting up the camera system and developing a cost-effective approach to monitoring the initial contrail stage. The paper is generally well-written and clear. However, I feel that sections 2.4, 3.1 and 3.2, which discuss contrail occurrence frequencies and differences between observations and models, are somewhat detailed. In contrast, section 3.3, which addresses the relationship between contrail occurrence and meteorological conditions, is less detailed and only scratches the surface. This is regrettable, as a more comprehensive comparison would provide more substantial recommendations for improving the Cocip model. In its current form, the analyses are not sufficiently conclusive, and numerous questions remain unanswered. In my specific comments, I provide several recommendations for expanding this section. Moreover, some conclusions are drawn without sufficient substantiation.

I can recommend publication after major revision.

**Major specific comments**

As mentioned above, the current analyses leave too many open questions. I believe addressing the following questions/issues will make the study stronger.

1. It should be made clear in the abstract that the contrail-cirrus that you are able to track are not yet mature, and that the observations cannot

adequately constrain model estimates of climatically relevant contrail-cirrus that typically live much longer than 30 minutes. It is recommended that this limitation be mentioned in the abstract.

2. Contrails in Cocip are initialized after wake vortex break-up. Hence t=0 in the model refers to a contrail age of several minutes. Have you considered this temporal offset in the comparison with the observations? You have not mentioned this; hence I assume you disregarded this effect. As the contrail lifetimes in the analyses are relatively small, neglecting this offset probably plays a role. Furthermore, your large class of observed contrails with t<2 min cannot be compared to CoCip outputs. Moreover, you can't even obtain an estimate of the contrail altitude, which is used in your retrieval algorithm, from CoCiP.

3. Comparing the contrail width for t<2 min is not particularly conclusive, as the contrail spreading occurs over longer time scales. This early stage is not simulated with CoCip (see point 1). It would be of interest to obtain a PDF of NWP RHi and the Cocip-simulated contrail lifetimes for those cases where you observe short-lived contrails (t<2 min). An analogous survey could be conducted for the class 2min <t<10min. However, for the class t>10 min, it would not be as meaningful, as the observed lifetimes are likely a lower limit of the real lifetimes. (For further information, please refer to point 6 regarding a proposed redefinition of the lifetime classes.)

   The information content of Fig. 4 is dominated by the class of very short-lived contrails. It may be beneficial to conduct an a-priori analysis of these contrails, followed by their removal from the data set. This approach would enable a more focused analysis that considers only longer-living contrails.

4. It would be beneficial to ascertain whether advection outside of the domain or contrail dissolution is the limiting factor of contrail lifetime in the observations. This should be quantified for each lifetime class.

5. One possibly interesting analysis could be cast into a scatter plot of observed and simulated lifetimes (for all those observed contrails that dissolve inside the observed region).

6. The validation of the Cocip model comprises several components.

    a. Firstly, the NWP input data (RHi, T, vertical wind shear) are crucial for the contrail initialization.

    b. Secondly, the Cocip model physics are of significance for the spreading and dissolution of the contrails.

The initialization aspect is of particular importance in your study, given that you are dealing with very short-lived or at most young contrail-cirrus. Nevertheless, the second bullet point is not given sufficient attention in the current study. It would be beneficial to gain further insight into contrail physical processes through a comparison exercise (despite the lack of mature contrail-cirrus with lifetimes of several hours in your data set). This is pertinent to the issue raised in point 1. Contrail model physics determine the lifecycle and climate impact estimates by Cocip. Therefore, your observations should be used to test the Cocip model physics in more detail than presently done.

The introduction of three lifetime classes is a valuable addition to the study. However, it would be more logical to categorise all observations of contrails that dissolve before vortex break-up into a single class. This class can then be employed for the validation of the Cocip initialization and classes with larger lifetimes for the validation of the Cocip physics.

7. With regard to point 5b, it is recommended that time series of the width evolution for selected observed contrails be presented and compared to the CoCip-simulated width evolution. It would be beneficial to ascertain whether the observed spreading rates align with those observed in similar lidar measurements (https://doi.org/10.1029/95GL03549). A similar approach could be taken to assess the optical thickness. Is it feasible to determine the optical thickness from pictures?

8. In general, the fact that you observe the evolution of specific contrails is not exploited much despite this sentence in the conclusion ("*Ground-based cameras provide a cost-effective way to observe contrails, and unlike satellite imagery, their higher relative spatiotemporal resolution enables effective tracking of the formation*

*and evolution of young contrails.*"). This aspect should be more fully explored.

9. I wonder how Cocip can be applied with vastly different timesteps It is my understanding that the implementation of model physics is valid for certain time step ranges. Using time steps of 30-60 minutes or 40 seconds should play a large role in how and which contrail physics are implemented. Hence, it would be interesting to investigate how the contrail properties depend on the time step chosen. It would also be valuable to ascertain whether simulating contrail-cirrus over several hours yields similar results irrespective of the time step. If this is not the case, how relevant is the validation of Cocip with small dt for a validation of Cocip typically run with much larger time steps?

10. Regarding the last paragraph in the conclusion: Cocip is a simplified model to study contrail-cirrus evolution. Therefore, it may not be only the input to the model (in form of NWP data), but the contrail modelling within CoCip that could require improvements. This should not be overlooked in this outlook.

**Minor specific comments**

a. Section 2.2.3 overemphasizes the importance of this aspect of contrail initialization. It is likely that this section has been included in the manuscript because some of the co-authors were involved in developing this Cocip extension. However, for the contrail width, which is the only contrail quantity evaluated in this study, this aspect should not play a significant role. Conversely, the physics relevant to contrail spreading of young contrail-cirrus in Cocip is not described.

b. How is contrail width defined and evaluated in Cocip? Please include a proper definition.

c. You use the term "rate", which typically refers to a change per time unit. Wouldn't "fraction" better fit in your case?

d. Line 34: it would be appropriate to also cite GCM results.

e. Figure 5: The current plot style makes it difficult to extract relevant information from Figure 5. Furthermore, the paragraph on this plot is quite short. It is recommended that the plot be improved or removed.

f. In lines 256 to 259 you mention two possible reason for the underestimation of contrail width in Cocip. But this list is by no means

exhaustive and without further analysis, this is merely a hypothesis. Therefore, it is recommended that the corresponding paragraphs in the abstract and conclusion be removed unless corroboration of this statement can be provided.

g. Line 256 "These results are consistent with Schumann et al. (2013)": Could you please be more explicit? Did Schumann 2013 already show that contrail width is too small in CoCiP compared to observations?

h. Lines 201-204: it appears reasonable (and it is also convenient) to attribute the discrepancies solely to issues with the NWP input data. However, it would be fair to consider the potential shortcomings of Cocip that may also contribute to the discrepancies. For example, the usage as an offline model or other aspects.

i. Could you please explain explicitly what is meant by the "modelled background pixel intensity" in line 168?

j. Line 264: I believe saying Cocip simulates an ellipse in the horizontal plane is wrong. I guess what you want to say is that one of the principle axes lies in the horizontal plane.

**Technical corrections**

i. Fig. 6 caption mentions "false positive rate" whereas the corresponding text mentions "false negative rate". I believe the latter is correct term.

ii. In the abstract, you mention a reduction of 17.5%. Given the uncertainties, I would feel more comfortable rounding it to 15% or 20%.

iii. Line 243: remove "an"

iv. Line 200: In all other formulas of a similar nature, the term "Camera" is the first index and "Cocip" the second. However, in this instance, the order has been reversed.

v. I believe, u and v are not defined in the text (only in plot axes)

vi. Line 307: remove "3".

---

## Author Comment (AC1)

**Response to Reviewer Comments**

We express our gratitude to the reviewers for their detailed comments, which significantly improved the quality, clarity, and narrative of this manuscript.

The reviewer's remarks are *italicized*, while our responses are presented in normal text. Blue text is used to cite passages from the manuscript and to track the changes made from the original to the revised manuscript. References cited in the blue text can be found in the revised manuscript. Line numbers refer to the clean version of the revised manuscript. Additionally, we have re-ordered some of the comments to enhance the flow and readability of this response document.

**REFEREE 1 (RC1)**

**General comments**

*The paper reports on the installation of a ground-based camera to take pictures of contrails in the upper troposphere and on the derivation of contrail-relevant statistics and properties from such data. Furthermore, the observations are compared on a statistical basis with model estimates obtained from a simplified contrail plume model. I commend the authors for their efforts in setting up the camera system and developing a cost-effective approach to monitoring the initial contrail stage. The paper is generally well-written and clear. However, I feel that sections 2.4, 3.1 and 3.2, which discuss contrail occurrence frequencies and differences between observations and models, are somewhat detailed. In contrast, section 3.3, which addresses the relationship between contrail occurrence and meteorological conditions, is less detailed and only scratches the surface. This is regrettable, as a more comprehensive comparison would provide more substantial recommendations for improving the Cocip model. In its current form, the analyses are not sufficiently conclusive, and numerous questions remain unanswered. In my specific comments, I provide several recommendations for expanding this section. Moreover, some conclusions are drawn without sufficient substantiation. I can recommend publication after major revision.*

**Major specific comments**

*As mentioned above, the current analyses leave too many open questions. I believe addressing the following questions/issues will make the study stronger.*

1. *It should be made clear in the abstract that the contrail-cirrus that you are able to track are not yet mature, and that the observations cannot adequately constrain model estimates of climatically relevant contrail-cirrus that typically live much longer than 30 minutes. It is recommended that this limitation be mentioned in the abstract.*

   - Thank you. We agree with these suggestions and have made clear in the abstract that our observations and results are only applicable to young contrails (< 35 minutes) that were formed under clear sky conditions:

     o [Abstract: Lines 11 – 14] "Here, we developed a methodology to **use ground-based cameras for** track**ing** and analys**inge young** contrails **(< 35 minutes) formed under clear sky conditions**  **and** compar**ed** these observations against **reanalysis meteorology and**

simulations from the contrail cirrus prediction model (CoCiP) with actual flight trajectories."

2. *The validation of the Cocip model comprises several components:*

   a. *Firstly, the NWP input data (RHi, T, vertical wind shear) are crucial for the contrail initialization.*

   b. *Secondly, the Cocip model physics are of significance for the spreading and dissolution of the contrails.*

*The initialization aspect is of particular importance in your study, given that you are dealing with very short-lived or at most young contrail-cirrus. Nevertheless, the second bullet point is not given sufficient attention in the current study. It would be beneficial to gain further insight into contrail physical processes through a comparison exercise (despite the lack of mature contrail-cirrus with lifetimes of several hours in your data set). This is pertinent to the issue raised in point 1. Contrail model physics determine the lifecycle and climate impact estimates by Cocip. Therefore, your observations should be used to test the Cocip model physics in more detail than presently done. The introduction of three lifetime classes is a valuable addition to the study. However, it would be more logical to categorise all observations of contrails that dissolve before vortex break-up into a single class. This class can then be employed for the validation of the Cocip initialization and classes with larger lifetimes for the validation of the Cocip physics.*

- Thank you for these suggestions. To address this comment, we have re-run the contrail simulation using the most up-to-date CoCiP algorithm (pycontrails v0.52.2) and included further analysis of the observed versus simulated contrail properties. As a result, major revisions have been made to the methodology and results section.

- More specifically, we have re-structured the validation of the contrail simulation workflow into four main components, which also helped to improve the overall narrative of the manuscript:

  i. Comparison of the observed contrail formation with the Schmidt-Appleman Criterion (SAC), where contrails are assumed to form in the simulation when the ambient temperature ($T_{amb}$) is below the SAC threshold temperature ($T_{SAC}$),

  ii. Comparison of the observed contrail formation with CoCiP's definition of persistent contrail formation (i.e., post wake vortex ice water content exceeding $10^{-12}$ kg kg$^{-1}$),

  iii. Evaluation of the observed and simulated contrail lifetimes relative to the ambient meteorology provided by NWPs, and

  iv. Comparison of the temporal evolution of the observed contrail geometric width relative to those simulated by CoCiP.

- **Component (i)** serves as a pre-cursor step to CoCiP and is mainly influenced by the NWP temperature fields and aircraft performance estimates. **Components (ii) to (iv)** are affected by CoCiP's simplifications and model physics, and NWP uncertainties.

- The additional analyses made for **components (i) and (ii)** are described in detail in Comment 3. Notably, we also revised our notations in the confusion matrix from ($Y_{CoCiP}$ and $N_{CoCiP}$) to ($Y_{Sim=SAC}$ and $N_{Sim=SAC}$) when evaluating the observations

against the SAC, and ($Y_{Sim=CoCiP}$ and $N_{Sim=CoCiP}$) when comparing the observations against CoCiP's definition of persistent contrail formation.

- **Component (iii)** is addressed in Comments 4–6, while **Component (iv)** is addressed in Comments 7, 16, 17, and 34.

- With reference to the suggestion "*it would be more logical to categorise all observations of contrails that dissolve before vortex break-up into a single class*", we note that we cannot precisely determine the timing of vortex break-up from the video footage. However, we have added new plots that segment the waypoints into two groups, including contrails that sublimate before and after the wake vortex break-up, as determined by the simulation (See Comment 3 and Fig. 6 in the revised manuscript).

3. *Contrails in Cocip are initialized after wake vortex break-up. Hence t=0 in the model refers to a contrail age of several minutes. Have you considered this temporal offset in the comparison with the observations? You have not mentioned this; hence I assume you disregarded this effect. As the contrail lifetimes in the analyses are relatively small, neglecting this offset probably plays a role. Furthermore, your large class of observed contrails with t<2 min cannot be compared to CoCip outputs. Moreover, you can't even obtain an estimate of the contrail altitude, which is used in your retrieval algorithm, from CoCiP.*

    - Thank you for highlighting this issue. The reviewer is correct to point out that CoCiP assumes that the effects of the wake vortex downwash, which determines the contrail altitude and persistence, occur instantaneously at $t = 0$. We have revised the manuscript to more clearly describe the different steps used by the simulation to initialise contrail formation and any associated assumptions:

        o [Main text: Lines 127 – 138] "Contrails form when the ambient temperature ($T_{amb}$) at the flight waypoint is below the $T_{SAC}$ which is estimated by,

        $$T_{SAC}[\text{K}] = (273.15 - 46.46) + 9.43\ln(G - 0.053) + 0.72[\ln(G - 0.053)]^2, \quad (2)$$

        **where $G$ is the gradient of the mixing line in a temperature-humidity diagram,**

        $$G = \frac{\text{EI}_{H_2O}\, p_{amb}\, c_p\, R_1}{Q_{fuel}\,(1-\eta)\, R_0}. \quad\quad (3)$$

        **$\text{EI}_{H_2O}$ is the water vapour emissions index and assumed to be 1.237 kg kg$^{-1}$ for Jet A-1 (Gierens et al., 2016), $\eta$ is provided by the aircraft performance model (Section 2.2.1), $p_{amb}$ is the pressure altitude at each waypoint, $c_p$ is the isobaric heat capacity of dry air (1004 J kg$^{-1}$ K$^{-1}$), and $R_1$ (461.51 J kg$^{-1}$ K$^{-1}$) and $R_0$ (287.05 J kg$^{-1}$ K$^{-1}$) are the gas constant for water vapour and dry air respectively.**

        **T**wo successive waypoints that satisfy the SAC forms a contrail segment **that can either be short-lived or persistent (Schumann, 1996)** . A parametric wake vortex model is then used to simulate the wake vortex downwash (Holzapfel, 2003)**, of which CoCiP assumes that the**

**process is instantaneous and does not resolve the temporal evolution of the wake vortex (Schumann, 2012).**

**P**ersistent contrails **in CoCiP** are defined when their post-wake vortex ice water content (IWC) remains above $10^{-12}$ kg kg$^{-1}$.''

- We have not included a temporal offset in our analysis because the video footage does not allow us to visually pinpoint the exact time of wake vortex breakup. Instead, we now compare the observed contrail formation against: (i) the SAC, where contrails are assumed to form in the simulation when $T_{amb} < T_{SAC}$; and (ii) CoCiP's definition of persistent contrail formation (i.e., post wake vortex ice water content exceeding $10^{-12}$ kg kg$^{-1}$). Specifically, our new results suggest that the SAC could overestimate contrail formation, while CoCiP's definition tended to underestimate contrail formation likely because of CoCiP assumes that the wake vortex downwash occurs instantaneously without a temporal offset.

  - [Main text: Lines 233 – 241]:

Table 1: Summary statistics for each day when contrails were observed by the camera**. For each of the five days, the observed** contrail formation **from the video footage is compared with the two different definitions of contrail formation in the simulation, i.e., using the SAC ($T_{amb} < T_{SAC}$) and CoCiP's definition of persistent contrail formation (post wake vortex contrail IWC > $10^{-12}$ kg kg$^{-1}$)**~~predictive accuracy from CoCiP relative to the camera observations for each day. The notation Y_Camera indicates that the camera observed contrails forming at the flight waypoint, N_Camera indicates that no contrails at the flight waypoint were observed by the camera, Y_CoCiP indicates that the CoCiP simulation estimates the formation of contrails at the flight waypoint, while N_CoCiP indicates that the CoCiP simulation did not predict contrails forming at the flight waypoint.~~

[revised manuscript text omitted]

- Notably, to prevent confusion, we have also revised our notations in the confusion matrix from ($Y_{CoCiP}$ and $N_{CoCiP}$) to ($Y_{Sim=SAC}$ and $N_{Sim=SAC}$) when evaluating the observations against the SAC, and ($Y_{Sim=CoCiP}$ and $N_{Sim=CoCiP}$) when comparing the observations against CoCiP's definition of persistent contrail formation.

4. *Comparing the contrail width for t<2 min is not particularly conclusive, as the contrail spreading occurs over longer time scales. This early stage is not simulated with CoCip (see point ). It would be of interest to obtain a PDF of NWP RHi and the Cocip-simulated contrail lifetimes for those cases where you observe short-lived contrails (t<2 min). An analogous survey could be conducted for the class 2min <t<10min. However, for the class t>10 min, it would not be as meaningful, as the observed lifetimes are likely a lower limit of the real lifetimes. (For further information, please refer to point  regarding a proposed redefinition of the lifetime classes.) The information content of Fig. 4 is dominated by the class of very short-lived contrails. It may be beneficial to conduct an a-priori analysis of these contrails, followed by their removal from the data set. This approach would enable a more focused analysis that considers only longer-living contrails.*

  - Thank you for this suggestion. We have now updated Fig. 5 (now Fig. 7 in the revised manuscript) visualise the ERA5-derived meteorology (temperature and RHi) versus the observed contrail lifetime and discuss these results:
    - [Main text: Lines 294 – 297]:

[Figure]

"Figure 7:  **Evaluation** of the observed contrail lifetime  **relative to the ERA5-derived RHi (y-axis) and** the difference between the ambient temperature ($T_{amb}$) and SAC threshold temperature ($T_{SAC}$) **(x-axis) at the time of contrail formation. This analysis includes all**  **waypoints with observed contrails**  ($Y_{Camera}$ )."

- o [Main text: Lines 286 – 293]: "**For waypoints with $Y_{Camera}$, we compared their observed contrail lifetimes against the ERA5-derived meteorology at the time of formation (Fig. 7). Our analysis shows that: (i) 98% of these contrails met the SAC ($T_{amb} < T_{SAC}$) in the simulation; (ii) 78% of short-lived contrails with observed lifetime under 2 minutes were formed under ice sub-saturated conditions (RHi < 100%); and (iii) 75% of persistent contrails with observed lifetime exceeding 10 minutes were formed in ice supersaturated conditions (RHi > 100%). The gradual decline in agreement between observations and NWP estimates over longer time periods suggests that the ERA5-derived temperature fields are generally more accurate than the humidity fields, as noted in previous studies (Gierens et al., 2020; Reutter et al., 2020), thereby leading to more accurate predictions of contrail formation compared to contrail persistence.** ~~A further evaluation of these waypoints ($Y_{Camera}$) shows a weak negative correlation between $dT_{SAC}$ and the observed contrail lifetime (R=-0.168, as shown in Fig. 5). This finding is consistent with previous research, suggesting that contrails forming at lower temperatures tend to have a lower ice water content and smaller ice crystal radius which, in turn, can increase the contrail lifetime.~~"

- Given these results, we agree with the reviewer that the comparison between the observed and simulated contrail widths for waypoints with observed lifetimes < 2 minutes is not particularly conclusive. These waypoints have been removed in our updated results (see Comment 7).

- Additionally, the suggestion to compare between the observed and simulated contrail lifetime has been addressed in Comment 7.

5.  *It would be beneficial to ascertain whether advection outside of the domain or contrail dissolution is the limiting factor of contrail lifetime in the observations. This should be quantified for each lifetime class.*

-   Thank you for this suggestion. We have revised the manuscript which now outlines the limiting factors for the observed contrail lifetime:

    o   [Main text: Lines 190 – 195] "All  waypoints with $Y_{Camera}$ are further classified **into three categories** based on their observed **contrail** lifetimes,  **defined as the** duration during which the contrail is  **observed by** the camera: (i) short-lived contrails with  lifetimes of fewer than 2 minutes; (ii) contrails with  lifetimes of between 2 and 10 minutes; and (iii) persistent contrails with  lifetimes of least 10 minutes (World Meteorological Organization, 2017). **We note that the observed contrail lifetimes in our study is restricted by the contrail either advecting out of the camera's field of view (see Fig. A2), becoming too small or faint to be visible in the footage, or sublimating within the observation domain.**"

-   Additionally, we now quantify the fate of the observed contrails for each lifetime class, specifying whether they advected outside the observation domain or sublimated within it:

    o   [Main text: Lines 281 – 285] "Among the 9**3** unique waypoints with observed contrails ($Y_{Camera}$), 7**3.3**% of the**m**  are short-lived **with observed lifetimes of less than (<** 2 min**utes.** **Of these short-lived contrails, 99.3% of them either became too small to be tracked or sublimated within the camera's field of view, while 0.7% advected out of it. Contrails with observed lifetimes ranging**  between 2 and 10 minutes **made up 12.5% of the observations, with 36% of them drifting beyond the camera's field of view.** **The remaining** 1**4.2**% of **contrails had**  observed lifetimes exceeding 10 minutes**, of which 64% of them advected beyond the camera's field of view.**"

6.  *One possibly interesting analysis could be cast into a scatter plot of observed and simulated lifetimes (for all those observed contrails that dissolve inside the observed region).*

-   Thank you for this suggestion. We have now included a scatter plot comparing the observed versus the simulated contrail lifetime, along with a discussion on the factors that could contribute to these discrepancies. To enhance clarity, the data has been segmented into two groups: (i) contrails that advected out of the camera's field of view (advected out); and (ii) contrails that became too small or faint to be visible, as well as those that sublimated within the observation domain.

    o   [Main text: Lines 309 – 314]:

[Figure]

"**Figure 8: Comparison between the observed and simulated contrail lifetime for waypoints with true positive outcomes ($Y_{Camera}$ & $Y_{Sim=CoCiP}$). Observed contrails are categorised based on their final known position: circles represent contrails that either sublimated or became too small or faint within the observation domain; while stars indicate that the contrail drifted out of the observation domain and can no longer be tracked. The colorbar represents the corrected ERA5-derived RHi at the time of contrail formation. The simulated contrail lifetime in this plot is constrained to 35 minutes to align with the maximum observed contrail lifetime.**"

o [Main text: Lines 298 – 308] "**Fig. 8 shows a poor visual agreement between the observed and simulated contrail lifetime, with the simulation generally underpredicting contrail lifetime when the ERA5-derived RHi is below 100% and could overestimate it when the RHi exceeds 100%. Several known factors likely contribute to this mismatch. Firstly, the ERA5 HRES humidity fields are known to have limitations, which often produce weakly supersaturated RHi estimates (Agarwal et al., 2022; Reutter et al., 2020; Teoh et al., 2022a). Although corrections were applied to ensure that the ERA5-derived RHi distribution is consistent with in-situ measurements (Section 2.2.2), the RHi uncertainties remain large at the waypoint level (Teoh et al., 2024a). Secondly, the spatial resolution of the ERA5 HRES (0.25° longitude × 0.25° latitude ≈ 18 × 28 km) is insufficient to capture the sub-grid scale RHi variabilities (Wolf et al., 2024). Here, we do not evaluate the effects of sub-grid scale RHi variabilities because of the small study domain, where the camera's field of view fits within 10 grid boxes of the ERA5 HRES (Fig. A2), and the limited sample size (n = 942 for waypoints with $Y_{Camera}$ over 14 h). Thirdly, the maximum observed contrail lifetime can be capped by the contrail drifting out of the observation domain or becoming too small or faint to be tracked (Fig. 3a). Thirdly, the maximum observed contrail lifetime can be capped by the contrail drifting out of the field of view or becoming too small or faint to be tracked (Fig. 3a).**"

7.  *With regard to point 2b, it is recommended that time series of the width evolution for selected observed contrails be presented and compared to the CoCip-simulated width evolution. It would be beneficial to ascertain whether the observed spreading rates align with those observed in similar lidar measurements (https://doi.org/10.1029/95GL03549). A similar approach could be taken to assess the optical thickness. Is it feasible to determine the optical thickness from pictures?*

- Thank you for this suggestion. We have updated Fig. 7 (now Fig. 9 in the revised manuscript) to show the temporal evolution of the observed and simulated contrail geometric widths for waypoints with true positive cases ($Y_{Camera}$ & $Y_{Sim=CoCiP}$) and with observed lifetimes exceeding 2 minutes (see below). For each contrail waypoint, we note that the noise in observed geometric width over time can be attributed to the extraction of the observed contrail pixel width from the video footage and reverse camera transformation, as will be discussed in Comments 16 and 34.

    o [Main text: Lines 330 – 336]:

[Figure]

"Figure 9:  **Comparison** between the observed and simulated contrail geometric width for  waypoints with true positive cases ($Y_{Camera}$ & $Y_{Sim=CoCiP}$) **and with observed lifetimes exceeding 2 minutes. Panel (a) shows a parity plot between the observed and simulated widths at single point in time, with the black lines representing the temporal evolution of the contrail width for each waypoint. Panel (b) illustrates the difference between the observed and simulated geometric widths as a function of the observed contrail age, with individual lines representing the temporal evolution of each contrail waypoint. The observed contrail pixel width is converted to the observed geometric width using the reverse camera transformation model (see Section 2.3).**"

- From these results, we identify that the underestimation in simulated contrail width tends to be largest for contrails with observed lifetimes under 5 minutes. This underestimation is most likely due to contrail model simplifications in initialising the persistent contrail width, which do not account for wake vortex dynamics and ambient meteorology. These discussions have now been included in the revised manuscript:

  - [Main text: Lines 316 – 329] "Fig**ure**. 79 compares the **temporal evolution of the** observed **and** contrail pixel and geometric width relative to the simulated **contrail geometric** widths CoCiP outputs for 533**705** segments from all waypoints with true positive cases ($Y_{Camera}$ & $Y_{Sim=CoCiP}$) **and observed lifetimes greater than 2 minutes**. On **average,** ur findings, as assessed by the root mean square error (RMSE) metric, suggest that the simulated observed contrail **geometric** width**s** tends to be smaller than the observed pixel width (by -6.8 pixels) and geometric width (by -330 m). These results are **around 100 m smaller than the observed widths over the observed contrail lifetime, with the largest underestimations occurring within the first five minutes (-280 m, on average, Fig. 9b). The tendency to underestimate the simulated contrail widths can be attributed to several known factors** and could be caused by the: (i) potential underestimation of sub-grid scale **uncertainties in** wind shear and turbulent mixing**, where their sub-grid scale variabilities cannot be resolved from the spatiotemporal resolution of** in the ERA5 HRES **(Hoffmann et al., 2019; Paugam et al., 2010; Schumann et al., 2013**); (ii) **contrail model errors resulting from the use of simplified physics, such as the Gaussian plume assumption which may not adequately represent the contrail cross-sectional area (Jensen et al., 1998b; Sussmann and Gierens, 1999; Unterstrasser and Gierens, 2010), instantaneous wake vortex assumption, and the initialisation of persistent contrail width solely based on the aircraft wingspan, c.f. Eq. (2), without considering wake vortex dynamics and ambient meteorology (Lewellen and Lewellen, 2001; Schumann, 2012); and (iii)** CoCiP's assumption of a Gaussian plume when simulating the evolution of contrails (Schumann, 2012) **definition of the simulated contrail width (i.e., the length across the y-axis of a Gaussian plume), which is inherently shorter than the maximum possible observed contrail width (i.e., length across the major axis of an inclined ellipse). These factors are among those identified and may not be exhaustive.**"

- For contrails with observed lifetimes greater than 5 minutes, our data suggests that the growth rates of the observed (mean of 16.9 m/min) and simulated contrail geometric width (mean of 59.3 m/min) are on the lower end relative to with those derived from lidar measurements (between 18 and 140 m/min) (Freudenthaler et al., 1995). However, due to the limited number of data points ($n = 14$) and the significant noise in the observed contrail widths (see Comments 16 and 34), we have decided not to include this comparison in the revised manuscript.

- We also note that no study to date has determined the contrail optical thickness from ground-based cameras, and achieving this would likely require a more accurate radiometric calibration of the camera system. We have now highlighted this as a potential avenue for future research:

o [Main text: Lines 396 – 397] "Future work can build upon our research by**: (i) developing a methodology to estimate the contrail optical thickness from ground-based cameras;** …"

8. *In general, the fact that you observe the evolution of specific contrails is not exploited much despite this sentence in the conclusion ("Ground-based cameras provide a cost-effective way to observe contrails, and unlike satellite imagery, their higher relative spatiotemporal resolution enables effective tracking of the formation and evolution of young contrails."). This aspect should be more fully explored.*

   • Thank you for this comment. We note that Section 3.3 now includes a more detailed analysis of the contrail width evolution over time (see Comment 7).

   • We have also revised this statement and paragraph to better align with the scope of this study (below):

      o [Main text: Lines 364 – 370] " Ground-based cameras  **can** observe contrails **at a** higher  spatiotemporal resolution **than satellite imagery, making them potentially valuable for validating the early contrail lifecycle as simulated by contrail models** . In this study, we develop a methodology to  analyse contrail **formation, persistence, and their geometric widths** from ground-based video footage, and subsequently compare these observations with contrail simulations.  **Our** contrail observations consist of 14 h of video footage recorded on five different days at Imperial College London's South Kensington Campus**.** **T**he actual flight trajectories  intersect**ing** with the camera's field of view were obtained from ADS-B telemetry**, and contrails formed by these flights were simulated with CoCiP using historical meteorology from the ECMWF ERA5 HRES reanalysis**."

9. *I wonder how Cocip can be applied with vastly different timesteps It is my understanding that the implementation of model physics is valid for certain time step ranges. Using time steps of 30-60 minutes or 40 seconds should play a large role in how and which contrail physics are implemented. Hence, it would be interesting to investigate how the contrail properties depend on the time step chosen. It would also be valuable to ascertain whether simulating contrail-cirrus over several hours yields similar results irrespective of the time step. If this is not the case, how relevant is the validation of Cocip with small dt for a validation of Cocip typically run with much larger time steps?*

   • Previous studies utilising CoCiP have employed a wide range of model time steps, depending on the model application and available computational resources:

i. Schumann et al. (2015) used a 60-minute model time step because CoCiP was coupled with the Community Atmosphere Model (CAM), which operates on a 60-minute time step. Additionally, the global simulations, spanning over 20 years, were constrained by computational resources thereby making shorter time steps impractical.

ii. Several regional contrail simulations over Japan, Europe, and the North Atlantic used a 30-minute time step, as these simulations were performed locally on consumer-grade machines (Schumann et al., 2021; Teoh et al., 2020, 2022).

iii. Schumann & Graf (2013) selected a 15-minute time step to match the time resolution of both the air traffic dataset and satellite imagery.

iv. The most recent global contrail simulation, which utilized cloud-based computing, was able to reduce the model time step to 5 minutes due to fewer computational constraints (Teoh et al., 2024).

- To align with the maximum observed contrail lifetime, we evaluated the sensitivity of the simulated contrail widths to the CoCiP model time step for persistent contrails with lifetimes under 35 minutes. Each simulation run involved the same set of persistent contrails formed by 100 randomly selected flights. Our results (see figure below) show that reducing the model time step from 30 minutes to 1-minute increases the mean simulated contrail width by 7%. The smaller widths at longer time steps can likely be attributed to the contrail lifetime ending prematurely, for example, when ambient conditions in the next time step is not ice supersaturated (RHi < 100%). In contrast, a 1-minute time step allows contrails in the simulation to persist longer, leading to increased contrail coverage area and width. While time step error is one of the many sources of error, it is not the most dominant one.

[Figure]

- This discussion is not included in the revised manuscript because evaluating the sensitivity of CoCiP to various inputs and model parameters is beyond the scope of this study. We note that these analyses will be addressed in more detail in two separate and upcoming publications, i.e., Engberg et al. (2024) and Ulrich Schumann (personal communication).

10. *Regarding the last paragraph in the conclusion: Cocip is a simplified model to study contrail-cirrus evolution. Therefore, it may not be only the input to the model (in form of NWP data), but the contrail modelling within CoCip that could require improvements. This should not be overlooked in this outlook.*

- Thank you for this suggestion. This is an important avenue of future research should be emphasised. We have revised this paragraph to address this point:
  - [Main text: Lines 400 – 401] "Future work can build upon our research by: (i) …; **(iv) conducting a large-scale comparison between the observed and simulated contrails to establish benchmark datasets, which can be used to validate and improve the accuracy of contrail models and the humidity fields provided by NWP models;** and ...""

**Minor specific comments**

11. *Section 2.2.3 overemphasizes the importance of this aspect of contrail initialization. It is likely that this section has been included in the manuscript because some of the co-authors were involved in developing this Cocip extension. However, for the contrail width, which is the only contrail quantity evaluated in this study, this aspect should not play a significant role. Conversely, the physics relevant to contrail spreading of young contrail-cirrus in Cocip is not described.*

- Thank you for highlighting this. We fully agree with this comment and have made the following changes to focus on methodological details relevant for this study:
  - Section 2.2.3 has been shortened and combined with Section 2.2.1:

    [Main text: Lines 103 – 109] "The  Base of Aircraft Data Family 4.2 (BADA 4) aircraft performance model **(EUROCONTROL, 2016) is used** to estimate the: (i) fuel mass flow rate; (ii) change in aircraft mass, assuming that the initial aircraft mass at the first known waypoint is **set** to the nominal  mass  provided by BADA; **and** (iii) overall efficiency ($\eta$)**T**he aircraft-engine specific non-volatile particulate matter (nvPM) number emissions index (EI$_n$)**, which strongly influences the initial contrail ice crystal properties, is estimated by interpolating the** engine**-specific nvPM emissions profile from** the ICAO Aircraft Engine Emissions Databank (EDB) (EASA, 2021) **relative to the non-dimensional engine thrust settings**  (Teoh et al., 2020b). All flights are assumed to be powered by conventional Jet A-1 fuel."

  - Section 2.2.4 (now Section 2.2.3) has been extended to include methodological details on the initialisation and evolution of the simulated contrail dimensions (width and depth):

    [Main text: Lines 138 – 152] "Persistent contrails **in CoCiP** are defined when their post-wake vortex ice water content (IWC) remains above $10^{-12}$ kg kg$^{-1}$.

The persistent contrail width ($W$) and depth ($D$), defined as the dimensions along the y- and z-axis of a Gaussian plume, are initialised as,

$$W_{t=0} = \frac{\pi}{4} S_a, \tag{4}$$

$$D_{t=0} = 0.5 \times dZ_{max}, \tag{5}$$

where $S_a$ is the aircraft wingspan and $dZ_{max}$ is the maximum vertical displacement of the contrail mid-point after the wake vortex breakup.

The evolution of  **different contrail properties are** then simulated using a Runge-Kutta scheme with  model time step**s** (d$t$) of 40 s. **More specifically, the change in contrail dimensions over time are estimated as,**

$$W_t = \sqrt{8\sigma_{yy}}, \tag{4}$$

$$D_t = \sqrt{8\sigma_{zz}}, \tag{5}$$

**where σ is a dispersion matrix that captures the spread of the contrail plume along the y- and z-axes. σ is influenced by various factors such as wind shear, contrail segment length, diffusivity, and d$t$ (Schumann, 2012). CoCiP assumes that the contrail segment is sublimated when it's**  ice particle number concentration or optical depth drops below $10^3$ m$^{-3}$ and $10^{-6}$, respectively, or when the mid-point of the contrail plume advects beyond the **simulation** domain  (40 – 60° N and 10° W – 10° E)."

*12. How is contrail width defined and evaluated in Cocip? Please include a proper definition.*

- Thank you for this suggestion. After reviewing the original CoCiP paper (Schumann, 2012), we can confirm that the contrail width is defined as the length across the y-axis of a Gaussian plume.

- For clarity improvements, we have revised the manuscript to include the definition of the contrail width in CoCiP:

  o [Main text: Lines 139 – 144] "**The persistent contrail width ($W$) and depth ($D$), defined as the dimensions along the y- and z-axis of a Gaussian plume, are initialised as,**

  $$W_{t=0} = \frac{\pi}{4} S_a, \tag{4}$$

  $$D_{t=0} = 0.5 \times dZ_{max}, \tag{5}$$

  **where $S_a$ is the aircraft wingspan and $dZ_{max}$ is the maximum vertical displacement of the contrail mid-point after the wake vortex breakup.**"

*13. You use the term "rate", which typically refers to a change per time unit. Wouldn't "fraction" better fit in your case?*

- Thank you for this suggestion. The term "rate" was originally used to be consistent with the standard statistical terminology used in a Confusion Matrix (Source: https://en.wikipedia.org/wiki/Confusion_matrix). However, we have made

significant revisions to the results section (see Comment 2), and the updated analysis and discussion no longer include these terms.

*14. Line 34: it would be appropriate to also cite GCM results.*

- Thank you for this suggestion. We have made several modifications to improve the clarity of this sentence and now include the most up-to-date GCM estimates from Märkl et al. (2024):

  - [Main text: Lines 38 – 41] "Recent studies suggests that the global annual mean contrail cirrus net radiative forcing (RF) in **2018 and** 2019 (**best-estimate of between 61 and 72** mW m$^{-2}$ **across three studies**) (**Märkl et al., 2024;** Quaas et al., 2021; Teoh et al., **2024a**) could be around two times greater than the RF from aviation's cumulative $CO_2$ emissions (34.3 [31, 38] mW m$^{-2}$ **at a 95% confidence interval**) (Lee et al., 2021)"

*15. Figure 5: The current plot style makes it difficult to extract relevant information from Figure 5. Furthermore, the paragraph on this plot is quite short. It is recommended that the plot be improved or removed.*

- Thank you. We agree with this comment and have updated this figure accordingly. The specific changes made are detailed in Comment 4.

*16. In lines 256 to 259 you mention two possible reasons for the underestimation of contrail width in Cocip. But this list is by no means exhaustive and without further analysis, this is merely a hypothesis. Therefore, it is recommended that the corresponding paragraphs in the abstract and conclusion be removed unless corroboration of this statement can be provided.*

- Thank you for your feedback. The identified factors contributing to an underestimation of the simulated contrail width are supported by existing studies and not merely hypothetical. For example, numerical simulations have assessed the influence of sub-grid scale variability in wind shear and turbulent mixing on the contrail plume dynamics and Lagrangian transport (Hoffmann et al., 2019; Paugam et al., 2010). Furthermore, studies using lidar measurements and large eddy simulations have shown that the Gaussian plume assumption may not adequately represent the contrail cross-sectional area as observed under real world conditions (Jensen et al., 1998; Sussmann and Gierens, 1999; Unterstrasser and Gierens, 2010).

- To address this point, we have revised the discussion in Section 3.3 and included these references. These changes should improve its clarity and broaden the likely factors that contribute to an underestimation of the simulated contrail width:

  - [Main text: Lines 319 – 329] "**The tendency to underestimate the simulated contrail widths is consistent with Schumann et al. (2013) and can be attributed to several known factors, including**: (i)  **uncertainties in** wind shear and turbulent mixing**, where their sub-grid scale variabilities cannot be resolved from the spatiotemporal resolution of**  the ERA5 HRES (**Hoffmann et al., 2019; Paugam et al., 2010; Schumann et al., 2013**); (ii) **contrail model errors resulting from the use of simplified physics, such as the Gaussian**

**plume assumption which may not adequately represent the contrail cross-sectional area (Jensen et al., 1998b; Sussmann and Gierens, 1999; Unterstrasser and Gierens, 2010), instantaneous wake vortex assumption, and the initialisation of persistent contrail width solely based on the aircraft wingspan, c.f. Eq. (2), without considering wake vortex dynamics and ambient meteorology (Lewellen and Lewellen, 2001; Schumann, 2012); and (iii)** CoCiP's  **definition of the simulated contrail width (i.e., the length across the y-axis of a Gaussian plume), which is inherently shorter than the maximum possible observed contrail width (i.e., length across the major axis of an inclined ellipse). These factors are among those identified and may not be exhaustive.**"

- Additionally, we also list several known and potential factors that contribute to discrepancies between the observed and simulated contrail formation and lifetime. We now acknowledge that these factors may not be exhaustive and have clarified this throughout the revised manuscript:

    o [Abstract: Lines 22 – 24] "**D**iscrepanc**ies** **between the observed and simulated contrail formation, lifetime and widths can be associated with**  **uncertainties in reanalysis meteorology due to known model limitations and sub-grid scale variabilities, contrail model simplifications, uncertainties in aircraft performance estimates, and observational challenges, among other possible factors** ."

    o [Main text: Lines 223 – 228] "**Section 3.1 compares the observed contrail formation with those predicted by the SAC and CoCiP. Section 3.2 evaluates the observed contrail lifetime against the ERA5-derived meteorology and simulated contrail lifetime, while Section 3.3 compares the temporal evolution of contrail width between the observation and simulation. Finally, Section 3.4 briefly explores the potential limitations in detecting contrails from the video footage. Across these sections, we discuss the known and potential factors that may contribute to the discrepancies between the observed and simulated contrail properties, while acknowledging that the list of factors may not be exhaustive.**"

    o [Main text: Lines 385 – 390] "**Overall, our results show a gradual decline in agreement between observations and simulations, particularly as contrails progress from formation to persistence. Discrepancies between the observed and simulated contrail properties stem from multiple sources including: (i) uncertainties in the ERA5 HRES humidity fields; (ii) sub-grid scale variabilities that cannot be captured by the spatiotemporal resolution of existing NWP models; (iii) contrail model assumptions and simplifications; (iv) uncertainties in the simulated aircraft overall efficiency, which influences $T_{SAC}$; (v) observational challenges (Fig. 3); and (vi) potentially other unidentified factors.**"

17. *Line 256 "These results are consistent with Schumann et al. (2013)": Could you please be more explicit? Did Schumann 2013 already show that contrail width is too small in CoCiP compared to observations?*

   - Thank you for highlighting this. We have made the following changes to address this point:

     o [Main text: Lines 317 – 320] "**On average,**  the simulated contrail **geometric** width ~~tends to be smaller than the observed pixel width (by ~6.8 pixels) and geometric width (by ~330 m)se results arecould be caused by the: (i) potential underestimation of sub-grid scale wind shear and turbulent mixing in the ERA5 HRES; and (ii) CoCiP's assumption of a Gaussian plume when simulating the evolution of contrails (Schumann, 2012).~~"

18. *Lines 201-204: it appears reasonable (and it is also convenient) to attribute the discrepancies solely to issues with the NWP input data. However, it would be fair to consider the potential shortcomings of Cocip that may also contribute to the discrepancies. For example, the usage as an offline model or other aspects.*

   - Thank you for this suggestion. We have made significant revisions to the results section (see Comments 2 and 3), where the paragraph mentioned by the reviewer has been replaced.

   - As noted in Comment 16, we now included "aircraft performance estimates" and "contrail model simplifications and assumptions" as contributing factors to discrepancies between the observed and simulated contrail formation. We also provide specific examples of the model processes that may contribute to these discrepancies (see Comment 3).

19. *Could you please explain explicitly what is meant by the "modelled background pixel intensity" in line 168?*

   - Thank you for this suggestion. We have modified the terminology slightly from "modelled background pixel intensity" to "estimated background pixel intensity", and made the following changes in the revised manuscript to address this point:

     o [Main text: Lines 209 – 213] "Instead, we compare the relative difference between the local pixel intensity ($P_{u,v}$) and the **estimated**  background pixel intensity ($\hat{P}_{u,v}^{B}$), **i.e., the estimated pixel intensity of the background sky assuming that the contrail is absent**,
     $$\Delta P_{u,v} = P_{u,v} - \hat{P}_{u,v}^{B}\text{,}$$ (8)

     **Here,** $\hat{P}_{u,v}^{B}$**, represented by the black line of best fit in the RGB plot of Fig. 4,** is **estimated**  using a Huber regression instead of a traditional least squares regression to minimise the regression sensitivity to outliers (Pedregosa et al., 2012)."

*20. Line 264: I believe saying Cocip simulates an ellipse in the horizontal plane is wrong. I guess what you want to say is that one of the principle axes lies in the horizontal plane.*

- Thank you for identifying this mistake. We have made the following changes in the manuscript for clarity improvements:

    o [Main text: Lines 319 – 329] "**The tendency to underestimate the simulated contrail width is consistent with** Schumann et al. (2013) **and can be attributed to several known factors, including** the: (i)  **uncertainties in** wind shear and turbulent mixing**, where their sub-grid scale variabilities cannot be resolved from the spatiotemporal resolution of** the ERA5 HRES **(Hoffmann et al., 2019; Paugam et al., 2010; Schumann et al., 2013);**  **(ii) contrail model errors resulting from the use of simplified physics, such as the Gaussian plume assumption which may not adequately represent the contrail cross-sectional area (Jensen et al., 1998b; Sussmann and Gierens, 1999; Unterstrasser and Gierens, 2010), instantaneous wake vortex assumption, and the initialisation of persistent contrail width solely based on the aircraft wingspan, c.f. Eq. (2), without considering wake vortex dynamics and ambient meteorology (Lewellen and Lewellen, 2001; Schumann, 2012); and (iii) CoCiP's definition of the simulated contrail width (i.e., the length across the y-axis of a Gaussian plume), which is inherently shorter than the maximum possible observed contrail width (i.e., length across the major axis of an inclined ellipse).**"

    o [Main text: Lines 337 – 345] "~~A visual comparison shows that the agreement between the observed and simulated contrail geometric width (Fig. 7b) is lower than the pixel width (Fig. 7a). The higher relative agreement between the observed and simulated contrail pixel width is partially explained by its dependence on the contrail-camera distance, i.e., contrails further away have a smaller pixel width, which can be estimated with high accuracy. In contrast,~~ **In addition to errors in the simulated contrail width, independent error sources in the observed contrail widths also contribute to the poor visual agreement between the observed and simulated contrail widths (Fig. 9a). Firstly, the presence of other contrails and natural cirrus can affect the Huber regression used to identify the contrail edges, c.f., Eq. (9), thereby contributing to errors in the observed contrail pixel width (Fig. 3b). Secondly, converting**  the observed **pixel width to** geometric width **introduces additional errors due to the lack of data on**  the: (i)  contrail altitude**, which we assume that the observed** is equal to the simulated contrail altitude in CoCiP (Section 2.4); and (ii) **inclination angle of the elliptical contrail plume, where parallax errors can contribute to a larger variability in the observed geometric width relative to the pixel width** . Assumption (i) is subject to uncertainties in the actual aircraft mass **and local meteorology**, **which can** result in additional errors when simulating the contrail vertical displacement caused by the wake vortex downwash."

**Technical corrections**

*21. Fig. 6 caption mentions "false positive rate" whereas the corresponding text mentions "false negative rate". I believe the latter is correct term.*

- Thank you for identifying this error. We note that significant revisions were made to the results section (see Comment 2) where Fig. 6 is no longer included in the revised manuscript.

*22. In the abstract, you mention a reduction of 17.5%. Given the uncertainties, I would feel more comfortable rounding it to 15% or 20%.*

- Thank you. We have revised this sentence in the abstract to incorporate the changes made in response to Comment 7:
  - [Abstract: Lines 20 – 21] "On average, the simulated contrail **geometric** width is **around 100 m**  smaller than the observed **(visible)**  width **over its observed lifetime, with the mean underestimation reaching up to 280 m within the first five minutes.**"

*23. Line 243: remove "an"*

- Thank you for identifying this error. We note that significant revisions were made to the results section (see Comment 2) where Fig. 6 is no longer included in the revised manuscript.

*24. Line 200: In all other formulas of a similar nature, the term "Camera" is the first index and "Cocip" the second. However, in this instance, the order has been reversed.*

- Thank you for highlighting this inconsistency. As mentioned in Comment 2, we have revised our notations in the confusion matrix from ($Y_{CoCiP}$ and $N_{CoCiP}$) to ($Y_{Sim=SAC}$ and $N_{Sim=SAC}$) when evaluating the observations against the SAC, and ($Y_{Sim=CoCiP}$ and $N_{Sim=CoCiP}$) when comparing the observations against CoCiP's definition of persistent contrail formation. We have also ensured that "Camera" and "Sim" are consistently used as the first and second indices, respectively.

*25. I believe, u and v are not defined in the text (only in plot axes)*

- Thank you. We have checked Section 2.3 and can confirm that the notations $u$ and $v$ have been defined in the text:
  - [Main text: Lines 159 – 164] "After correcting for distortions, we project the simulated contrail waypoints and dimensions onto the video footage using a camera transformation model **that** follows a two-step process: (i) the real-world 3D positions (i.e., ADS-B flight waypoints and the simulated mid-point and edges of the contrail plumes) are mapped to a 3D camera coordinate system ($X$, $Y$, $Z$) using an extrinsic (rotation) matrix; followed by (ii) transforming the 3D camera coordinates ($X$, $Y$, $Z$) to a 2D pixel coordinate system ($u$, $v$) using an intrinsic (camera) matrix."

- Additionally, we have also renamed the x- and y-axes in Figure 8 (now Figure 3 in the revised manuscript) to be consistent with Figure 2 and Figure A1:
    - x-axis "Pixel x-coordinate, $u$"
    - y-axis "Pixel y-coordinate, $v$"

*26. Line 307: remove "3".*

- Thank you for identifying this type error. It has been removed.

**REFEREE 2 (RC2)**

*This paper presents an analysis of surface-based RGB video camera images to detect contrails over London during 5 different days. The advantage of using the ground observations is that narrower and fainter contrails not detected from most satellite imagery can be observed and tracked through the early stages of development providing data for comparison with contrail prediction/diagnostic models. In this paper, the approach is limited to otherwise clear skies and comparisons are made to contrails predicted by widely used CoCiP model informed by adjusted ERA 5 reanalysis input. From the 283 flights, the paper finds that in 69% of the cases, the CoCiP correctly predicts the presence or absence of contrails with a low false alarm rate complemented by a much higher false negative rate resulting in a definitive underestimate of contrail formation by the model. However, the false negatives are more prominent for short-lived contrails and at higher temperatures (lower altitudes), where the ambient temperature and $T_{SAC}$ are close. The CoCiP appears to perform best for persistent contrails at low temperatures. Further comparisons show that the CoCiP tends to underestimate the contrail width by 15%, on average. Errors in formation and width by the model are presumed to be due mainly to assumptions in the model and to temperature and humidity uncertainties in the adjusted ERA 5 profiles, specifically sub-grid scale variability. Future work is proposed to expand the network of cameras, apply the analysis methodology to many more flights, and combine ground and satellite measurements to fully track persistent contrails.*

**General comments**

*This is essentially a demonstration study that shows how ground-based cameras can be used to assess contrail formation and provide some guidance for modelling the same. It is very limited in actual sampling (only 14 hours in 5 days) and in the range of atmospheric conditions. Satellite analyses indicate that only 15% of persistent contrails occur in otherwise clear skies [Bedka et al., GRL, 2013]. While it proposes to expand the use of cameras in future studies, no way forward past the clear-sky limitation is proffered. Perhaps, that is not a problem as it could be valuable for examining contrails for that clear-sky portion of the natural cloudiness spectrum. While the sampling limitations are mentioned at the end, they should be emphasized more both for the cloud conditions and the number of cases. I would recommend publication after that large concern and others highlighted below are addressed.*

**Specific comments**

*27. Line 103.   How were the ERA5 humidity fields corrected? Does the IAGOS dataset correspond to the same times or is it an average correction applied generally?  That needs*

*to be fleshed out, even if it is in one of the references. What were the magnitudes of the alterations?*

- Thank you for these suggestions. We have revised the manuscript to incorporate these suggestions:

  o [Main text: Lines 116 – 125] "We **apply the humidity correction from Teoh et al. (2022a)**  to ensure that the **ERA5-derived** RHi **has a probability density function that**  is consistent with in-situ measurements from the In-service Aircraft for a Global Observing System (IAGOS) dataset (Petzold et al., 2015; Boulanger et al., 2022),

$$\mathbf{RHi_{corrected}} = \begin{cases} \dfrac{\mathbf{RHi}}{\mathbf{a_{opt}}} & \textbf{for } \left(\dfrac{\mathbf{RHi}}{\mathbf{a_{opt}}}\right) \leq 1 \\[2em] \mathbf{min}\left(\left(\dfrac{\mathbf{RHi}}{\mathbf{a_{opt}}}\right)^{\mathbf{b_{opt}}}, \mathbf{RHi_{max}}\right) & \textbf{for } \left(\dfrac{\mathbf{RHi}}{\mathbf{a_{opt}}}\right) > 1 \end{cases} \tag{1}$$

  **where $\mathbf{RHi_{max}} = 1.65$, $\mathbf{a_{opt}} = 0.9779$ and $\mathbf{b_{opt}} = 1.635$. Eq. (1) is expected to be applicable to this study because its coefficients were calibrated using RHi measurements over the North Atlantic (40 – 75° N and 50 – 10° W), which corresponds to the same latitude band as our study domain (40 – 60° N and 10° W – 10° E). While Eq. (1) improves the goodness of fit between the measured and ERA5-derived RHi distribution and corrects for average biases (Teoh et al., 2022a), we note that it does not correct for the RHi errors at specific waypoints (Teoh et al., 2024a). Thus, RHi uncertainties at each waypoint can remain significant.**"

28. *Fig. 2 Were the wide contrails in top right and lower right of (a) and (b) not picked up by CoCiP? If so, it should be noted in the text.*

- After reviewing the earlier footage, we confirmed that these contrails were formed outside the camera's observation domain and subsequently drifted into view. Although the simulation domain spans ±10° in both longitude and latitude from the camera's location (as noted in Section 2.2), these specific contrails were not picked up by the contrail simulation.

- This has now been highlighted in the figure caption of the revised manuscript:

  o [Main text: Lines 166 – 171] "Figure 2: Example of the flight trajectories and simulated contrail  **dimensions** from CoCiP that are superimposed **on**to the video footage using the camera transformation model **(detailed in Section 2.3)**. The flight trajectories and contrails were observed on 5-Nov-2021 between 09:16:40 and 09:22:40 (UTC). **Note that the persistent contrails visible in the top right and lower right of panels (a) and (b) were formed outside the observation domain and subsequently drifted into the camera's field of view, and the absence of labels on these contrails suggests that they were most likely false negative outcomes ($\mathbf{Y_{Camera}}$ & $\mathbf{N_{Sim=CoCiP}}$).**"

*29. Line 202. Perhaps, the sub grid scale variability effect could be estimated by assessing the frequency of correct predictions within the same grid box.*

- We thank the reviewer for this suggestion. However, an evaluation of the sub-grid scale variability effects with our dataset may not yield meaningful results due to the following limitations:

    i. **Small study domain**: The spatial distribution of the observed contrails (51 – 52°N and 0.25°W – 1°E, now included as Fig. A2 in the revised manuscript) fits within 10 grid boxes of the ERA5 HRES (0.25° longitude × 0.25° latitude), and

    ii. **Limited sample size**: The number of contrail waypoints observed by the camera is relatively small (n = 942), which is further distributed over 14 hours and across 6 pressure levels (i.e., 150, 175, 200, 225, 250 and 300 hPa).

- We note that major revisions have been made to the results section (see Comment 2), and this discussion has been moved from Section 3.1 (Contrail formation) to Section 3.2 (Contrail lifetime). Additionally, we have also included a short discussion to mention the above point in the revised manuscript:

    o [Main text: Lines 298 – 307] "**Fig. 8 shows a poor visual agreement between the observed and simulated contrail lifetime, with the simulation generally underpredicting contrail lifetime when the ERA5-derived RHi is below 100% and could overestimate it when the RHi exceeds 100%. Several known factors likely contribute to this mismatch. Firstly, the ERA5 HRES humidity fields are known to have limitations, which often produce weakly supersaturated RHi estimates (Agarwal et al., 2022; Reutter et al., 2020; Teoh et al., 2022a). Although corrections were applied to ensure that the ERA5-derived RHi distribution is consistent with in-situ measurements (Section 2.2.2), the RHi uncertainties remain large at the waypoint level (Teoh et al., 2024a). Secondly, the spatial resolution of the ERA5 HRES (0.25° longitude × 0.25° latitude ≈ 18 × 28 km) is insufficient to capture the sub-grid scale RHi variabilities (Wolf et al., 2024). Here, we do not evaluate the effects of sub-grid scale RHi variabilities because of the small study domain, where the camera's field of view fits within 10 grid boxes of the ERA5 HRES (Fig. A2), and the limited sample size (n = 942 for waypoints with Y$_{Camera}$ over 14 h).**"

    o [Appendix: Lines 425 – 428]

[Figure]

"**Figure A2: Location of the camera (51.4988°N, 0.1788°W) and the spatial distribution of observed contrails within its field of view (n = 942 for waypoints with Y$_{Camera}$). The grid boxes represent the spatial resolution of the ERA5 HRES (0.25° longitude × 0.25° latitude).**"

30. *Line 205.  It would be useful to examine the sensitivity of the CoCiP predictions to perturbations in the ERA 5 humidity fields in this paper. That may help determine how far off the RHi is in the ERA5, providing evidence for future corrections and possibly improve the CoCiP accuracy for these cases.*

    - Thank you for this suggestion. Previous studies have explored the:
        i. sensitivity of the simulated contrail properties (from CoCiP) to the corrections applied to the ERA5 humidity fields (Schumann et al., 2021; Teoh et al., 2022, 2024), and
        ii. the accuracy of the ERA5 humidity fields compared to in-situ measurements from aircraft sensors and radiosondes (Agarwal et al., 2022; Reutter et al., 2020; Teoh et al., 2024).

    - In the revised manuscript, we have partially addressed this question by:
        i. Using the observed contrail lifetime from the ground-based camera to evaluate the ERA5's capability to predict short-lived contrail formation (which depends on the temperature fields) and contrail persistence (which depends on the humidity fields) (see Comment 4), and
        ii. comparing the observed and simulated contrail lifetime and associated their differences to the ambient RHi (see Comment 6).

    - However, due to the small sample size (14 hours of video footage and 1,582 flight waypoints) and the lack of observational data for the ambient temperature and RHi, we recommend that future research use a more comprehensive dataset to further investigate this issue. This has now been mentioned in the conclusions section as an avenue for further research:
        - [Main text: Lines 400 – 401] "Future work can build upon our research by: (i) ……; **(iv) conducting a large-scale comparison between the observed and simulated contrails to establish benchmark datasets, which can be used to validate and improve the accuracy of contrail models and the humidity fields provided by NWP models;** and ..."

31. *Line 205. Would we expect the humidity fields to be more accurate over this domain than say the middle of the Atlantic, since much of the area includes surface sites where radiosonde profiles taken? If so, would we expect lower detection accuracies in other areas where radiosonde profiles are not available for assimilation into the ERA5? Or, does the assimilation process damp out the impact of the more accurate data?*

    - Humidity fields in the ERA5 HRES are assimilated using measurements from multiple sources, including satellites, radiosondes and aircraft sensors (Hersbach et al., 2020).

    - Although no study has specifically assessed the difference in RHi accuracy between regions at the same latitude (i.e., UK vs. North Atlantic), we hypothesize that any potential improvements in the accuracy of initial conditions due to increased

observations may be offset by the simplified assumptions used in the ECMWF Integrated Forecast System (IFS). For example, the IFS simulates the evolution of humidity fields with a relaxation time (i.e., time required for excess humidity to be deposited into ambient particles and reach equilibrium) currently set to one model time step in grid boxes with natural cirrus presence (Tompkins et al., 2007).

- Teoh et al. (2024) compared the ERA5-derived RHi with in-situ RHi measurements from the In-service Aircraft for a Global Observing System (IAGOS) dataset and identified a latitude-dependent bias. Specifically, the ERA5-derived ISSR coverage (grid cells with RHi > 100%) could be overpredicted in the tropics and subtropics (0 – 40°N) and underpredicted at higher latitudes (above 40°N). The causes of these discrepancies are not yet fully understood but could be due to: (i) differing physical processes controlling the upper troposphere humidity; or (ii) variations in the density and frequency of radiosonde observations across latitudes.

- We note that the humidity correction that was applied in our study domain was calibrated based on IAGOS RHi measurements over the North Atlantic. Since both regions are located at comparable latitudes, the corrected RHi fields in our simulation should account for the latitude-dependent biases identified by Teoh et al. (2024). The revised manuscript now clarifies this point, along with the limitations of the humidity correction:

  - [Main text: Lines 120 – 125] "**Eq. (1) is expected to be applicable to this study because its coefficients were calibrated using RHi measurements over the North Atlantic (40 – 75° N and 50 – 10° W), which corresponds to the same latitude band as our study domain (40 – 60° N and 10° W – 10° E). While Eq. (1) improves the goodness of fit between the measured and ERA5-derived RHi distribution and corrects for average biases (Teoh et al., 2022a), we note that it does not correct for the RHi errors at specific waypoints (Teoh et al., 2024a). Thus, RHi uncertainties at each waypoint can remain significant.**"

32. *Table 1. How independent are the samples from the 1,619 unique waypoints? I would expect many of those points to be in the same air mass on a given day. For example, the accuracy is very low on 14 Jan compared to the other days.*

- The summary statistics listed in Table 1 depend on the ambient weather conditions of each day, so the samples are not independent. Upon further analysis, we identified that the very low accuracy on 14-Jan-2022 is due to contrails forming in ice-subsaturated air masses (RHi < 100). This has now been noted in the revised manuscript:

  - [Main text: Lines 271 – 273] "**Notably, on 14-Jan-2022, correct contrail predictions dropped sharply from 83.8% to 42.9%, with no persistent contrails were predicted in the simulation, because the ERA5-derived RHi at all waypoints were well below ice supersaturation (0.07–0.79, Fig. 6).**"

- Additionally, we have also revised Table 1 to include their mean difference between the ambient and SAC threshold temperature ($dT_{SAC} = T_{amb} - T_{SAC}$) and RHi for each day (see Comment 4). Fig. 6 in the revised manuscript (below) plots the ERA5-

derived d$T_{SAC}$ and RHi for all waypoints across five different days (represented by different colors), which clearly shows that the samples are not independent:

- o [Main text: Lines 274 – 279]

[Figure]

"**Figure 6: Corrected RHi from the ERA5 HRES versus the difference between the ambient ($T_{amb}$) and SAC threshold temperature ($T_{SAC}$) for all waypoints across five days: (a) with; and (b) without contrails observed from the video footage. In both plots, data points with no fill (circles) represent waypoints where contrails did not form in the simulation ($N_{Sim=SAC}$), crosses indicate waypoints that satisfied the SAC in the simulation ($Y_{Sim=SAC}$), and filled data points denote waypoints where persistent contrails were formed in the simulation ($Y_{Sim=CoCiP}$).**"

33. *Fig. 6. Y-axis should be False negative rate? Same in the caption.*

- Thank you for identifying this error. We note that significant revisions were made to the results section (see Comment 2) where Fig. 6 is no longer included in the revised manuscript.

34. *Lines 254-259 & Fig. 7. There should be some discussion of the distribution of the points rather than just stating the average difference. In Fig. 7b, there is a significant number of samples near the measured zero-line that have much more spreading than in the prediction, while the others are clustered at or above the agreement line. Can you shed some light on this? One particular day or type of contrail?*

- Thank you for this suggestion. We have updated Fig. 7 (now Fig. 9 in the revised manuscript) to show the individual data points and their temporal evolution in contrail width (see Comment 7). The significant number of data points near the zero-line in the original figure was caused by false positive waypoints ($N_{Camera}$ & $Y_{Sim=CoCiP}$), which have been removed in the updated figure.

- Additionally, we have revised the manuscript to provide a more in-depth discussion on the potential factors causing the simulated contrail width to be underestimated relative to observations (see Comment 16). We also address other independent error sources that influence the accuracy of the observed contrail geometric width (see below):

- o [Main text: Lines 337 – 348] "~~A visual comparison shows that the agreement between the observed and simulated contrail geometric width (Fig. 7b) is lower than the pixel width (Fig. 7a). The higher relative agreement between the observed and simulated contrail pixel width is partially explained by its dependence on the contrail-camera distance, i.e., contrails further away have a smaller pixel width, which can be estimated with high accuracy.~~ **In addition to errors in the simulated contrail width, independent error sources in the observed contrail widths also contribute to the poor visual agreement between the observed and simulated contrail widths (Fig. 9a). Firstly, the presence of other contrails and natural cirrus can affect the Huber regression used to identify the contrail edges, c.f. Eq. (9), thereby contributing to errors in the observed contrail pixel width (Fig. 3b). Secondly, converting the observed pixel width to geometric width introduces additional errors due to the lack of data on**  the: (i) **contrail altitude, which we assume that the observed**  contrail altitude is equal to the simulated contrail altitude in CoCiP (Section 2.4); and (ii)  **inclination angle of the elliptical contrail plume, where parallax errors can contribute to a larger variability in the observed geometric width relative to the pixel width**. Assumption (i) is subject to uncertainties in the actual aircraft mass **and local meteorology**, **which can** result in additional errors when simulating the contrail vertical displacement caused by the wake vortex downwash  **We** evaluate  the sensitivity of the observed  geometric width **to these factors** by varying the assumed contrail altitude **and the altitude at one of the contrail edges** by ± 100 m . Our results indicate that  **the inclination angle** has a significantly greater influence on the observed contrail geometric width (± 36%)  **compared** to **the altitude** assumption  (± 0.9%)."

35. *Line 275-279 & Fig. 8b. It is stated that contrails predicted on 5 Nov "appear to show a reasonable agreement." That may be stretching it a bit. There is one contrail that agrees near the bottom center and another predicted contrail that occurs within an observed one, but it is not clear which observed contrail the top-right prediction is supposed to be matched up with. Moreover, at least half of the observed contrails are not even predicted. Different wording may be more appropriate.*

*We should point to that the image only shows contrails that were formed inside the camera image. So non-labelled observed contrails do not mean they have not been predicted.*

- Thank you for highlighting this. We agree with these points and have revised the manuscript accordingly (below). Additionally, we have also re-ordered the figures in the revised manuscript, so what was previously Fig. 8 is now Fig. 3 in the revised manuscript.
  - o [Main text: Lines 172 – 177] "Figure **3**: Examples of **the simulated** contrails that were initially formed outside the **camera's observation**  domain

and subsequently **drifted**  into  view on : (a) **9- Nov-2021 at 10:02:40 UTC; and (b)** 5-Nov-2021 at 09:09:20 UTC. **The CoCiP-simulated contrail dimensions are superimposed onto the video footage using the camera transformation model (detailed in Section 2.3). Note that the absence of labels on some of the observed contrails in panel (b) indicates that they were most likely false negative outcomes ($Y_{Camera}$ & $N_{Sim=CoCiP}$).**"

- [Main text: Lines 350 – 362] "We  **visually** examined contrails that  initially formed outside the  **observation** domain and were subsequently advected into  view, where the results yielded mixed outcomes. **Firstly,**  on **5-Nov-2021**  at 09:09:20 UTC**, some predicted contrails aligned well with the**  observations (Fig. **3**b). **However, not all observed contrails were predicted by the model, and there were notable differences in the locations of predicted and observed contrails. We note that contrail-contrail and cloud-contrail overlapping further complicated the identification of contrail edges and the extraction of contrail widths.**

**Secondly**  on **9-Nov-2021**  at 10:02:40 UTC, **we were unable to visually confirm the presence of contrails in the video footage (Fig. 3a), despite** the simulation predict**ing** contrail cirrus with a mean optical depth of 0.024 [0.002, 0.056] (5th and 95th percentile)**. This** suggest that these contrails **could be misclassified as false positive cases ($N_{Camera}$ & $Y_{Sim=CoCiP}$) because their optical depths were**  below **or**  close to the lower visibility  threshold **limit** for ground-based observers (optical depth of $< 0.02$) (Kärcher et al., 2009).  **Although** faint white grains were visible in the video footage (Fig. **3**a), **it remains challenging to determine**  whether these features represent contrail cirrus**.** **This difficulty underscores the challenges that remote sensing methods, including ground-based cameras, have**  **with** detecting optically thin contrails below a **yet-to-be determined** threshold optical depth **(Driver et al., 2024; Mannstein et al., 2010; Meijer et al., 2022)** ."

36. *Conclusions: Please note the discussion in the general comments about future use of this approach. Also, it would be useful in the future to add a cloud lidar to the analysis to enable evaluation of the some of the error sources that contribute to disagreements in this paper.*

- Thank you. We acknowledge the following feedback made in the general comments and above:

  i. *"While it proposes to expand the use of cameras in future studies, no way forward past the clear-sky limitation is proffered."*

  ii. *"While the sampling limitations are mentioned at the end, they should be emphasized more both for the cloud conditions and the number of cases."*

iii.    *"It would be useful in the future to add a cloud lidar to the analysis to enable evaluation of the some of the error sources that contribute to disagreements in this paper"*

- Feedback (i) and (iii) can be addressed by combining the camera observations with satellite imagery and lidar measurements. This point has now been included in the conclusions in the revised manuscript:

  - [Main text: Lines 396 – 403] "Future work can build upon our research by**: (i) developing a methodology to estimate the contrail optical thickness from ground-based cameras;** (**ii**) establishing a network of ground-based cameras to observe contrails across a larger set of flights and over a wider domain**, while also mitigating the sensitivity of**  camera model**s**  to contrail altitude; (iii) combining ground-based (**e.g., cameras and lidars**) and satellite observations to track the whole contrail lifecycle **and beyond cloud free conditions** ; **(iv) conducting a large-scale comparison between the observed and simulated contrails to establish benchmark datasets, which can be used to validate and improve the accuracy of contrail models and the humidity fields provided by NWP models;** and (v) integrating ground-based observations with contrail forecasts, thereby reducing the uncertainties in the real-time decision making processes for flight diversions to minimise the formation of strongly warming contrails.

- For feedback (ii), we now highlight the sampling limitations in the abstract and conclusions:

  - [Abstract: Lines 11 – 14] "Here, we develop**ed** a methodology to **use ground-based cameras for** track**ing** and analys**ing**  **young** contrails (**< 35 minutes**) **formed under clear sky conditions** , **and** compar**ed**  these observations against **reanalysis weather data and** simulations from the contrail cirrus prediction model (CoCiP) with actual flight trajectories."

  - [Main text: Lines 371 – 372] "In total, **we identified** 1,**582**  **flight** waypoints from 28**1**  flights  from the video footage**, with contrails observed in 60% of these waypoints (Y$_{Camera}$) under clear sky conditions** ."

---

## Referee Report (RR1)

Review of

**Ground-based contrail observations: comparisons with flight telemetry and contrail model estimates**

by **Low et al.** submitted to **AMT**

**Review of the revised manuscript (iteration #1)**

**General comments**

I acknowledge the efforts to substantially revise selected parts of the manuscript. Moreover, I appreciate the detailed point-to-point replies. However, I believe that a few aspects still require improvement. Several plots are quite fuzzy and some are not adequately described in the legend/caption or text body. Furthermore, the analysis does not fully exploit the observational data set. I believe that more conclusive results could be achieved with little additional effort.

**Major specific comments**

1. Contrail persistence
   A more conclusive analysis and comparison between observation and simulation could be achieved. In lines 190–195, three lifetime categories for contrail observations are mentioned. However, the analysis does not make significant use of the 2-10 min and >10 min categories.

2. Table 1
   In Table 1, CoCiP persistence is compared to "Camera observes contrail," but a more accurate comparison would be to "Camera observes contrail with lifetime > 2 min". It is recommended that a third block be added to the table using "$Y_{Camera>2min}$", which hopefully improves the agreement with CoCiP.

3. Figs. 2 & 3
   The intention behind showing Figs. 2 and 3 is good and the plots should be kept in the manuscript. However, the description, explanation and interpretation are left to the readers. A single sentence is insufficient to convey the full meaning of the plots;

additional clarification is necessary ("Figures 2 and 3 provide examples of the superimposed flight trajectories and/or simulated contrail properties to the video footage.").

Moreover, please ensure that the legend lists only items that do appear in the plots. The inserted text is readily legible. It would be preferable to produce plots with enhanced quality and to focus on the content that is intended to be conveyed. For instance, it is not evident why multiple black lines intersecting the contrails have been plotted. I do not see the added value of plotting all the black lines (I do not motivate them either).

4. Figs. 5 & 6

   I am unable to understand the right panel of Fig. 5. The title indicates that only true positives are displayed, yet the legend lists all four combinations

   In my opinion, the information content of the present plots is not overly high as most aspects are straightforward to interpret. For instance, the fact that all grey and blue symbols in Fig.5 are to the right of the vertical line, while the green and red ones are to the left. Fig. 5 and 6 only uses the binary information whether or not contrail formation was observed. I strongly recommend that you show analogous plots for observed lifetimes > 2 min and > 10min, which can be compared to similar CoCip categories.

   As previously stated in my review of the original submission, it is recommended that the fact that individual contrails are observed over time be exploited to a greater extent.

   This is the excerpt from the previous review round:

   "In general, I realize that you do not really exploit the fact that you observe the evolution of specific contrails despite this sentence in the conclusion ("*Ground-based cameras provide a cost-effective way to observe contrails, and unlike satellite imagery, their higher relative spatiotemporal resolution enables effective tracking of the formation and evolution of young contrails.*"). This should be better exploited.

**Minor specific comments**

I.  Line 40: Märkl et al (2024) focuses on measurements and the climate impact of SAF contrails. The Bier & Burkhardt (2022) paper, which

you cite a few lines below, would be a better reference, as it deals with classical contrails from kerosene and the main topic of the paper is about GCM results.

II. Around line 60: Iwabuchi et al (2012) is worth to be mentioned.

III. Fig. 4: the two red lines in the picture are not explained. Moreover, DAL73 is not explained. Would it suffice to just draw the one black line for the intersection that is depicted in the left panel?

IV. Explanation of Table 1 starting from line 263: It would be worth mentioning that $Y_{Sim=CoCiP}$ cases are a subset of the $Y_{Sim=SAC}$ cases. Hence, it is trivial that the values in the first two lines of the CoCiP block are smaller than the analogous entries the SAC block. Likewise, the values in the third and fourth line are smaller in the CoCiP block.

V. Fig. 8: I appreciate that you mention a "poor visual agreement", which is indeed the case. Nevertheless, I suggest to spend a few more lines on describing on what can be seen in the plot (cases with y=0, y=35min or x=0). Currently, the plot is described in only two lines 298-300, before starting with the plot interpretation in line 300 spanning over many lines.

VI. Fig. 9: In the figure caption, you mention that the black lines represent the "temporal evolution …". These are only the thin black lines. The thick black line is the 1:1 line.

VII. Lines 305-306: For me, an analysis using a smaller study is even more affected by sub-grid scale variations. Hence, "because of the small study domain" sounds a bit awkward. I would have expected "despite of…"

**Technical corrections**

i. Line 29: reaches -> exceeds?
ii. Line 127: remove ","
iii. Line 151: its
iv. Line 343: " ."
v. Line 360: Missing full stop.

---

## Author Response (AR2)

**Response to Reviewer Comments**

We thank the referees for reviewing the revised manuscript. Their detailed comments further improved the quality, clarity, and narrative of this manuscript.

The reviewer's remarks are *italicized*, while our responses are presented in normal text. Blue text is used to cite passages from the manuscript and to track the changes made from the original to the revised manuscript. References cited in the blue text can be found in the revised manuscript. Line numbers refer to the clean version of the revised manuscript.

**REFEREE 1 (RC1)**

**General comments**

*I acknowledge the efforts to substantially revise selected parts of the manuscript. Moreover, I appreciate the detailed point-to-point replies. However, I believe that a few aspects still require improvement. Several plots are quite fuzzy and some are not adequately described in the legend/caption or text body. Furthermore, the analysis does not fully exploit the observational data set. I believe that more conclusive results could be achieved with little additional effort.*

**Major specific comments**

1. *Contrail persistence: A more conclusive analysis and comparison between observation and simulation could be achieved. In lines 190–195, three lifetime categories for contrail observations are mentioned. However, the analysis does not make significant use of the 2-10 min and >10 min categories.*

   - Thank you for this feedback. We have now utilised these lifetime categories in two additional analyses as suggested by the reviewer (see Comments 2 and 5).

2. *Table 1: In Table 1, CoCiP persistence is compared to "Camera observes contrail," but a more accurate comparison would be to "Camera observes contrail with lifetime > 2 min". It is recommended that a third block be added to the table using "YCamera>2min", which hopefully improves the agreement with CoCiP.*

   - Thank you for this suggestion. After further consideration, we believe this additional analysis is more suitable for inclusion in the text of Section 3.1 rather than in Table 1. The reason is that filtering the waypoints to "$Y_{Camera} > 2$ minutes" would reduce the evaluation to only two outcomes ($Y_{Camera}$ & $Y_{Sim,}$ and $Y_{Camera}$ & $N_{Sim}$), whereas Table 1 presents four different outcomes ($Y_{Camera}$ & $Y_{Sim}$; $Y_{Camera}$ & $N_{Sim}$; $N_{Camera}$ & $Y_{Sim}$; and $N_{Camera}$ & $N_{Sim}$). This creates an inconsistency. Additionally, applying a lifetime-based filter would also reduce the sample size that were previously listed in Table 1.

   - We have now included this analysis in the revised manuscript (below). Our results align with the reviewer's hypothesis, showing that the simulation becomes more capable of predicting contrail formation when the observed contrail lifetime is greater than two minutes ($Y_{Camera} > 2$ minutes):

     o [Main text: Lines 273 – 285] "Unlike with the SAC, the percentage of false negative waypoints ($Y_{Camera}$ & $N_{Sim=CoCiP} = 21.2\%$) is nearly four times

higher than the false positive waypoints ($N_{Camera}$ & $Y_{Sim=CoCiP}$ = 5.7%) (c.f. $Y_{Camera}$ & $N_{Sim=SAC}$ = 1.1% vs. $N_{Camera}$ & $Y_{Sim=SAC}$ = 23.1%). **False negative waypoints also tend to occur at lower altitudes (35100 ± 2600 feet at 1σ) and at sub-saturated RHi conditions (0.68 ± 0.19 at 1σ) relative to those with true positive outcomes (37500 ± 2700 feet and 1.02 ± 0.29) (Fig. 5b). Notably, on 14-Jan-2022, correct contrail predictions dropped sharply from 83.8% to 42.9%, with no persistent contrails predicted in the simulation, because the ERA5-derived RHi at all waypoints were well below ice supersaturation (0.07–0.79, Fig. 6).** The **difference in accuracy between the SAC and CoCiP's definition**  of persistent contrail formation is most likely due to contrail model simplifications (**i.e.,** instantaneous **wake vortex downwash**) which can underestimate the simulated contrail lifetime**s, particularly**  for short-lived contrails. **Indeed, when waypoints are segmented by the observed contrail lifetime, the simulation correctly predicted contrail formation for only 55% of waypoints with short-lived contrails ($Y_{Camera}$ < 2 minutes & $Y_{Sim=CoCiP}$). However, correct predictions increased significantly to 96% for waypoints with observed lifetimes between 2 and 10 minutes, and to 86% for waypoints with observed contrails persisting beyond 10 minutes.**"

- o [Abstract: Lines 16 – 17] "**When evaluating contrails with observed lifetimes of at least 2 minutes, the simulation's correct prediction rate for contrail formation increases to over 85%.**"

- o [Conclusions: Lines 401 – 403] "**When waypoints with $Y_{Camera}$ are segmented based on their observed contrail lifetime, the simulation accurately predicted contrail formation for only 55% of short-lived contrails ($Y_{Camera}$ < 2 minutes & $Y_{Sim=CoCiP}$), while correct predictions rose to over 85% for contrails with observed lifetimes exceeding 2 minutes ($Y_{Camera}$ ≥ 2 minutes & $Y_{Sim=CoCiP}$).**"

3. _Figs. 2 & 3: The intention behind showing Figs. 2 and 3 is good and the plots should be kept in the manuscript. However, the description, explanation and interpretation are left to the readers. A single sentence is insufficient to convey the full meaning of the plots; additional clarification is necessary ("Figures 2 and 3 provide examples of the superimposed flight trajectories and/or simulated contrail properties to the video footage.")._

   _Moreover, please ensure that the legend lists only items that do appear in the plots. The inserted text is readily legible. It would be preferable to produce plots with enhanced quality and to focus on the content that is intended to be conveyed. For instance, it is not evident why multiple black lines intersecting the contrails have been plotted. I do not see the added value of plotting all the black lines (I do not motivate them either)._

   - Thank you for this feedback, we have made the following changes in the revised manuscript to better describe Figures 2 and 3:

      - o [Main text: Lines 159 – 168] "After correcting for distortions, we project the **ADS-B**  waypoints and **simulated contrail** dimensions onto the video footage using a camera transformation model that

follows a two-step process**. First,:**  the real-world 3D positions (i.e., ADS-B  waypoints and the simulated mid-point and edges of the contrail plumes) are mapped **in**to a 3D camera coordinate system (*X, Y, Z*) using an extrinsic (rotation) matrix**. Next,**;  the 3D camera coordinates (*X, Y, Z*) **are transformed in**to a 2D pixel coordinate system (*u, v*) using an intrinsic (camera) matrix. **Using this two-step process, Fig. 2 shows the ADS-B waypoints and simulated contrails superimposed onto the video footage, specifically young contrails less than 6 minutes old that were formed within the camera's field of view. Similarly, Fig. 3 projects the simulated dimensions of aged contrails (i.e., those initially formed outside the camera's field of view and subsequently advected into it) onto the footage and compares them with the observed contrails.** Further details of the camera transformation model can be found in Appendix A3. "

- Additionally, we have re-plotted Figures 2 and 3 to enhance the image resolution.

- In Figure 2, the updated sub-plots now focus specifically on the flight trajectory and contrails formed by a single flight (callsign "UAL31"), rather than multiple flights, to improve clarity. We also removed: (i) items in the legend that were not visible in the sub-plots; and (ii) the black lines perpendicular to the contrail waypoints, which were previously used to sample the RGB pixel intensity and estimate the observed contrail widths (refer to Figure 4 in the revised manuscript).

  o [Main text: Lines 169 – 172] Updated Figure 2:

[Figure]

"Figure 2: Example of the flight trajector**y** and simulated contrail dimensions from **the flight with callsign "UAL31", both of which**

 are superimposed onto the video footage using the camera transformation model (detailed in Section 2.3). The flight trajector**y** and **persistent** contrails were observed on 5-Nov-2021 between 09:**37:20** and 09:**45:20** (UTC). ~~Note that the persistent contrails visible in the top right and lower right of panels (a) and (b) were formed outside the observation domain and subsequently drifted into the camera's field of view, and the absence of labels on these contrails suggests that they were most likely false negative outcomes (Y~Camera~ & N~Sim=CoCiP~).~~"

- ○ [Main text: Lines 173 – 179] Updated Figure 3:

[Figure]

"Figure 3: Examples of the simulated contrails that were initially formed outside the camera's observation domain and subsequently drifted into view

on: (a) 9-Nov-2021 at 10:02:40 UTC; and (b) 5-Nov-2021 at 09:09:20 UTC. The CoCiP-simulated contrail dimensions are superimposed onto the video footage using the camera transformation model (detailed in Section 2.3).  **In panel (a), the faint signals and absence of observed contrails suggest that they could be false positive outcomes ($N_{Camera}$ & $Y_{Sim=CoCiP}$). In panel (b),** the absence of labels on some  observed contrails  indicates that they were most likely false negative outcomes ($Y_{Camera}$ & $N_{Sim=CoCiP}$)."

4. _Fig. 5: I am unable to understand the right panel of Fig. 5. The title indicates that only true positives are displayed, yet the legend lists all four combinations. In my opinion, the information content of the present plots is not overly high as most aspects are straightforward to interpret. For instance, the fact that all grey and blue symbols in Fig.5 are to the right of the vertical line, while the green and red ones are to the left._

- Thank you for identifying the mistake in Fig. 5b. We want to clarify that the data points included in Fig. 5b filter only for waypoints that satisfied the SAC ($Y_{Sim=SAC}$), rather than true positive outcomes ($Y_{Camera}$ & $Y_{Sim=SAC}$). Additionally, we note that the data points presented in the updated Fig. 5b differ slightly from the earlier version due to a minor bug in our code, where we inadvertently filtered for data points with $Y_{Camera}$ **or** $Y_{Sim=SAC}$.

- We have also updated the caption of Fig. 5 to clarify the definitions of false negative and true negative outcomes in panel (b):

  - [Main text: Lines 263 – 268] "Figure 5: Joint plot of the aircraft barometric altitude versus the: (a) difference between the ambient ($T_{amb}$) and SAC threshold temperature ($T_{SAC}$) across all flight waypoints; and (b)  corrected RHi from the ERA5 HRES for waypoints that satisfy the SAC in the simulation  ($\text{\sout{$Y_{Camera}$ &}}$ $Y_{Sim=SAC}$). In both **panels** , green data points represent true positive outcomes ($Y_{Camera}$ & $Y_{Sim}$), red for false positive outcomes ($N_{Camera}$ & $Y_{Sim}$), blue for false negative outcomes ($Y_{Camera}$ & $N_{Sim}$), and grey for true negative outcomes ($N_{Camera}$ & $N_{Sim}$). **In panel (b), the false negative ($Y_{Camera}$ & $N_{Sim=CoCiP}$) and true negative outcomes ($N_{Camera}$ & $N_{Sim=CoCiP}$) correspond to waypoints that satisfied the SAC in the simulation but did not persist beyond the wake vortex phase.**"

- While we agree with the reviewer that most aspects in Fig. 5 are straightforward to interpret – specifically that all grey and blue symbols are positioned to the right of the vertical line, while green and red symbols are to the left – we have included it to emphasise two key takeaways from this figure:

  i. False positive ($N_{Camera}$ & $Y_{Sim=SAC}$) and false negative outcomes ($Y_{Camera}$ & $N_{Sim=SAC}$) occur closer to threshold conditions ($T_{amb} \approx T_{SAC}$ and RHi $\approx$ 100%) compared to true positives ($Y_{Camera}$ & $Y_{Sim=SAC}$) and true negatives ($N_{Camera}$ & $N_{Sim=SAC}$), and

  ii. False negative waypoints ($Y_{Camera}$ & $N_{Sim=CoCiP}$) also tend to occur at lower altitudes (35100 ± 2600 feet at 1σ) relative to those with true positive waypoints ($Y_{Camera}$ & $Y_{Sim=CoCiP}$) (37500 ± 2700 feet).

These points are already discussed in the manuscript (see Lines 258 – 260 and Lines 275 – 277).

5. _Figs. 5 & 6: Fig. 5 and 6 only uses the binary information whether or not contrail formation was observed. I strongly recommend that you show analogous plots for observed lifetimes > 2 min and > 10min, which can be compared to similar CoCip categories. As previously stated in my review of the original submission, it is recommended that the fact that individual contrails are observed over time be exploited to a greater extent._

_This is the excerpt from the previous review round: "In general, I realize that you do not really exploit the fact that you observe the evolution of specific contrails despite this sentence in the conclusion ("Ground-based cameras provide a cost-effective way to observe contrails, and unlike satellite imagery, their higher relative spatiotemporal resolution enables effective tracking of the formation and evolution of young contrails."). This should be better exploited._

- Thank you for this feedback. We note that the observed contrail lifetimes have been compared with the ERA5-derived RHi and temperature at the point and time of their formation, albeit on a continuous spectrum rather than using the three lifetime categories (see Fig. 7 in the revised manuscript).

- To address this comment, we have now included additional analysis in the revised manuscript that utilises the three lifetime categories (< 2 minutes, 2 – 10 minutes, and > 10 minutes):

  o [Main text: Lines 305 – 310] Updated Figure 7:

[Figure]

"Figure 7: Evaluation of the observed contrail lifetime relative to the ERA5-derived **meteorology at the point and time of their formation for all waypoints with observed contrails ($Y_{Camera}$). Panel (a) compares the observed contrail lifetime with the** RHi (y-axis) and the difference between the ambient temperature ($T_{amb}$) and SAC threshold temperature ($T_{SAC}$) (x-axis)**. Panel (b) shows the cumulative density functions of the initial RHi, with the data points segmented into three groups based on their observed contrail lifetimes, i.e., those lasting fewer than 2 minutes (gray), between 2 and 10 minutes (orange), and more than 10 minutes (red)** at the point and time of contrail formation. This analysis includes all waypoints with observed contrails ($Y_{Camera}$)."

- [Main text: Lines 299 – 304] "For waypoints with $Y_{Camera}$, we compared their observed contrail lifetimes against the ERA5-derived meteorology at the **point and** time of **their** formation (Fig. 7). Our analysis shows that: (i) 98% of the **observed** contrails  **fulfilled** the SAC ($T_{amb} < T_{SAC}$) in the simulation; (ii) 78% of short-lived contrails **($Y_{Camera}$ < 2 minutes)**  were formed under ice sub-saturated conditions (RHi < 100%)**, with a mean RHi of 81 ± 25% (1σ); (iii) 59% of contrails with observed lifetimes of between 2 and 10 minutes also formed under ice sub-saturated conditions, but the mean RHi is higher at 103 ± 32%**; and (iv) 75% of persistent contrails **($Y_{Camera}$ > 10 minutes)**  were formed in ice supersaturated conditions (RHi > 100%)**, with a mean RHi of 124 ± 26%**."

    - [Conclusions: Lines 403 – 406] "Notably, **among the waypoints with $Y_{Camera}$: (i)** 98% of **them**  fulfilled the SAC**; (ii)** 78% of  short-lived contrails (observed lifetimes < 2 minutes) initially formed at RHi < 100%**; (iii) 59% of contrails with observed lifetimes ranging between 2 and 10 minutes also formed at RHi < 100%; while (iv)**  75% of persistent contrails (observed lifetimes > 10 minutes) formed at RHi > 100% (Fig. 7)."

- In addition to the analyses mentioned here and in Comment 2, we also note that the contrail evolution over time has also been exploited when comparing the observed and simulated contrail geometric width in Section 3.3 (Fig. 9). However, this analysis was conducted on a continuous spectrum and did not necessitate the use of three lifetime categories.

**Minor specific comments**

6. *Line 40: Märkl et al (2024) focuses on measurements and the climate impact of SAF contrails. The Bier & Burkhardt (2022) paper, which you cite a few lines below, would be a better reference, as it deals with classical contrails from kerosene and the main topic of the paper is about GCM results.*

    - Thank you for this suggestion. Both studies (Bier & Burkhardt, 2022; Märkl et al., 2024) use the same modelling approach to simulate contrails globally. We agree that Bier & Burkhardt (2022) may be a more relevant reference, as it focuses specifically on contrails formed by conventional jet fuel. We initially selected Märkl et al. (2024) because: (i) Bier & Burkhardt (2022) only reports the global annual mean contrail net radiative forcing for 2006; while (ii) Märkl et al. (2024) provides more recent estimates (i.e., 2018 global contrail net RF), which aligns with the 2018-2019 period mentioned in this sentence. Nevertheless, we also acknowledge that Märkl et al. (2024) derives its 2018 estimates by scaling air traffic activity from 2006 to 2018 levels.

    - To address this comment, we have decided to include Bier & Burkhardt (2022) alongside Märkl et al. (2024), rather than replacing the latter reference in this sentence:

        - [Main text: Lines 38 – 41] "Recent studies suggest that the global annual mean contrail cirrus net radiative forcing (RF) in 2018 and 2019 (best-estimate of between 61 and 72 mW m$^{-2}$ across three studies) **(Bier and**

Burkhardt, 2022a; Märkl et al., 2024; Quaas et al., 2021; Teoh et al., 2024a) could be around two times greater than the RF from aviation's cumulative $CO_2$ emissions (34.3 [31, 38] mW m$^{-2}$ at a 95% confidence interval) (Lee et al., 2021)."

7. *Around line 60: Iwabuchi et al (2012) is worth to be mentioned.*

- Thank you. We agree with this suggestion and have included the Iwabuchi et al. (2012) reference in this sentence:

  o [Main text: Lines 57 – 62] "While satellite observations can partially address some limitations of in-situ measurements by enabling a large number of contrails to be measured, matched with specific flights and tracked over time (Duda et al., 2019; Gryspeerdt et al., 2024; **Iwabuchi et al., 2012;** Marjani et al., 2022; Tesche et al., 2016; Vázquez-Navarro et al., 2015), they still face challenges in detecting young contrails with sub-pixel width, aged contrail cirrus that has lost its line-shaped structure, instances of cloud-contrail overlap, and contrails with small optical depths (< 0.05) (Kärcher et al., 2010; Mannstein et al., 2010; Meijer et al., 2022)."

8. *Fig. 4: the two red lines in the picture are not explained. Moreover, DAL73 is not explained. Would it suffice to just draw the one black line for the intersection that is depicted in the left panel?*

- Thank you for this feedback. We have updated this figure to enhance the clarity of its legend in response to Comment 3, as well as revising the caption to address this comment:

  o [Main text: Lines 192 – 198] Updated Figure 4:

[Figure]

"Figure 4: Pixel colour intensity profiles of the contrail waypoint at Line 5 (shown at the bottom right **panel**). **The contrail observed in the bottom right panel was formed by the flight with callsign "DAL73".** **In the left panel, the black L**inear trendlines  **represent** the **best-fit** background colour intensity for each RGB channel. The solid  yellow **vertical**  line **marks**  the mid-point of the observed  contrail plume while the dashed (horizontal) yellow line indicates the estimated contrail pixel width. **In both**

> **the left and bottom right panels, the purple line indicates the centre of the simulated contrail plume from CoCiP, and in the bottom right panel, the red lines show the simulated contrail edges.**"

- After further consideration, we decided to retain the five black lines in the plot to demonstrate that the observed and simulated contrail widths were compared at each flight waypoint.

9. *Explanation of Table 1 starting from line 263: It would be worth mentioning that $Y_{Sim=CoCiP}$ cases are a subset of the $Y_{Sim=SAC}$ cases. Hence, it is trivial that the values in the first two lines of the CoCiP block are smaller than the analogous entries the SAC block. Likewise, the values in the third and fourth line are smaller in the CoCiP block.*

  - Thank you for this suggestion. We have made the following changes in the revised manuscript to address this comment:

    o [Main text: Lines 269 – 273] " CoCiP's defin**es** persistent contrail formation **as occurring when the**  post wake vortex contrail IWC **exceeds** $\geq 10^{-12}$ kg kg$^{-1}$ ($Y_{Sim=CoCiP}$),  **and adiabatic heating from the wake vortex downwash is assumed to occur instantaneously at the time of contrail initialisation. As a result, waypoints with $Y_{Sim=CoCiP}$ are a subset of $Y_{Sim=SAC}$. Using CoCiP's definition of persistent contrails,** the overall **accuracy of**  contrail predictions  **over** five days decreased slightly from 75.8% (SAC approach) to 73.1%, with significant variability between individual days (Table 1)."

10. *Fig. 8: I appreciate that you mention a "poor visual agreement", which is indeed the case. Nevertheless, I suggest to spend a few more lines on describing on what can be seen in the plot (cases with y=0, y=35min or x=0). Currently, the plot is described in only two lines 298-300, before starting with the plot interpretation in line 300 spanning over many lines.*

  - Thank you for this suggestion. We agree with this and have made the following changes in the revised manuscript:

    o [Main text: Lines 311 – 317] "Fig. 8 shows a poor visual agreement between the observed and simulated contrail lifetime, with the simulat**ed** **lifetimes being strongly influenced by the ERA5-derived RHi. Specifically, the simulation always predicts contrails with lifetimes below 5 minutes**  when the  RHi is  **less than** 100%**, often underestimating the observed contrail lifetimes. Additionally, the simulation consistently predicts contrails with lifetimes exceeding 2 minutes**  when the RHi  **is above** 100%**, even though around half of these waypoints were observed with short-lived contrails (< 2 minutes). It also tends to predict contrail lifetimes longer than 35 minutes when the RHi exceeds 120%, though evaluating these predictions is challenging as the maximum observed contrail lifetime can be limited by the contrail drifting out of the field of view or becoming too small or faint to be tracked (Fig. 3a).**"

- For data points where y = 35min, the maximum simulated contrail lifetime has been capped to 35 minutes to align with the longest observed contrail lifetime. This was previously noted in the caption of Figure 8.

11. *Fig. 9: In the figure caption, you mention that the black lines represent the "temporal evolution …". These are only the thin black lines. The thick black line is the 1:1 line.*

- Thank you identifying this mistake. We have made the following changes in the revised manuscript to rectify this:

    o [Main text: Lines 336 – 341] "Figure 9: Comparison between the observed and simulated contrail geometric width for waypoints with true positive cases **and with observed lifetimes exceeding 2 minutes** ($Y_{Camera} > 2$ minutes & $Y_{Sim=CoCiP}$) . Panel (a) shows a parity plot between the observed and simulated widths at single point in time, with the black line representing the **1:1 line** . Panel (b) illustrates the difference between the observed and simulated geometric widths as a function of the observed contrail age **For panels (a) and (b),** individual lines **connecting different data points** represent the temporal evolution of **the contrail width at** each contrail waypoint. The observed contrail pixel width is converted to the observed geometric width using the reverse camera transformation model (see Section 2.3)."

12. *Lines 305-306: For me, an analysis using a smaller study is even more affected by sub-grid scale variations. Hence, "because of the small study domain" sounds a bit awkward. I would have expected "despite of…"*

- Thank you. We have made the following changes in the revised manuscript to address this comment:

    o [Main text: Lines 321 – 326] "Secondly, the spatial resolution of the ERA5 HRES (0.25° longitude × 0.25° latitude ≈ 18 × 28 km) is insufficient to capture the sub-grid scale RHi variabilities **that have been observed from in-situ measurements** (Wolf et al., 2024).  **Given** the small study domain, where the camera's field of view fits within 10 grid boxes of the ERA5 HRES (Fig. A2), **our simulation would be particularly impacted by these sub-grid scale effects. However, we do not evaluate these effects due to our small**  sample size (n = 942 for waypoints with $Y_{Camera}$ **distributed** over 14 h **and 10 grid boxes**)."

**Technical corrections**

13. *Line 29: reaches -> exceeds?*

14. *Line 127: remove ","*

15. *Line 151: its*

16. *Line 343: " ."*

*17. Line 360: Missing full stop.*

- Thank you for identifying these technical errors, the necessary corrections have been applied to address Points 13 to 17.

**REFEREE 2 (RC2)**

*18. While the authors have made a good effort to acknowledge the limitations of this study in terms of representing only clear-sky contrails, they leave out any quantification of the representativeness. A good reference to that effect was provided in the original review, Bedka et al., Geophysical. Res. Lett., 2013. Given that there is knowledge available on this subject, the authors should inform their readers about how much of the contrail phenomenon this study addresses.*

- Thank you for highlighting this. We overlooked this comment in the original review. We have now incorporated the findings of Bedka et al. (2013), which estimate that contrails form under clear sky conditions only about 15% of the time:
  - [Main text: Lines 414 – 419] "Nevertheless, we acknowledge the potential limitations of our study, including the small sample size and an inherent bias toward selecting contrails formed under high-pressure systems (i.e., clear sky conditions), **which is estimated to account for only 15% of all contrails in the Northern Hemisphere (Bedka et al., 2013). This selection bias**  exclud **a significant portion of** contrails formed in low-pressure systems associated with storms or overcast weather.  **Such discrepancies in** synoptic weather conditions could introduce varying error patterns in NWP models, which may **propagate and affect**  the accuracy of the simulated contrail outputs."

**REFERENCES**

Bedka, S. T., Minnis, P., Duda, D. P., Chee, T. L., & Palikonda, R. (2013). Properties of linear contrails in the Northern Hemisphere derived from 2006 Aqua MODIS observations. *Geophysical Research Letters*, *40*(4), 772–777. https://doi.org/10.1029/2012GL054363

Bier, A., & Burkhardt, U. (2022). Impact of Parametrizing Microphysical Processes in the Jet and Vortex Phase on Contrail Cirrus Properties and Radiative Forcing. *Journal of Geophysical Research: Atmospheres*, *127*(23), e2022JD036677. https://doi.org/10.1029/2022JD036677

Iwabuchi, H., Yang, P., Liou, K. N., & Minnis, P. (2012). Physical and optical properties of persistent contrails: Climatology and interpretation. *Journal of Geophysical Research: Atmospheres*, *117*(D6). https://doi.org/10.1029/2011JD017020

Märkl, R. S., Voigt, C., Sauer, D., Dischl, R. K., Kaufmann, S., Harlaß, T., Hahn, V., Roiger, A., Weiß-Rehm, C., Burkhardt, U., Schumann, U., Marsing, A., Scheibe, M., Dörnbrack, A., Renard, C., Gauthier, M., Swann, P., Madden, P., Luff, D., … Le Clercq, P. (2024). Powering aircraft with 100 % sustainable aviation fuel reduces ice crystals in contrails. *Atmospheric Chemistry and Physics*, *24*(6), 3813–3837. https://doi.org/10.5194/ACP-24-3813-2024